

# Different expression pattern of flowering pathway genes contribute to male or female organ development during floral transition in the monoecious weed *Ambrosia artemisiifolia* L. (*Asteraceae*)

Kinga Klára Mátyás[1], Géza Hegedűs[2], János Taller[1], Eszter Farkas[1], Kincső Decsi[1], Barbara Kutasy[1], Nikoletta Kálmán[3], Erzsébet Nagy[1], Balázs Kolics[1] and Eszter Virág[1]

[1] Department of Plant Science and Biotechnology, University of Pannonia, Georgikon Faculty, Keszthely, Hungary
[2] Department of Economic Methodology, University of Pannonia, Georgikon Faculty, Keszthely, Hungary
[3] Department of Biochemistry and Medical Chemistry, University of Pecs Medical School, Szentagothai Research Center, Pecs, Hungary

Corresponding author
Eszter Virág, virag@georgikon.hu

## ABSTRACT

The highly allergenic and invasive weed *Ambrosia artemisiifolia* L. is a monoecius plant with separated male and female flowers. The genetic regulation of floral morphogenesis is a less understood field in the reproduction biology of this species. Therefore the objective of this work was to investigate the genetic control of sex determination during floral organogenesis. To this end, we performed a genome-wide transcriptional profiling of vegetative and generative tissues during the plant development comparing wild-growing and in vitro cultivated plants. RNA-seq on Illumina NextSeq 500 platform with an integrative bioinformatics analysis indicated differences in 80 floral gene expressions depending on photoperiodic and endogenous initial signals. Sex specificity of genes was validated based on RT-qPCR experiments. We found 11 and 16 uniquely expressed genes in female and male transcriptomes that were responsible particularly to maintain fertility and against abiotic stress. High gene expression of homologous such as FD, FT, TFL1 and CAL, SOC1, AP1 were characteristic to male and female floral meristems during organogenesis. Homologues transcripts of LFY and FLC were not found in the investigated generative and vegetative tissues. The repression of AP1 by TFL1 homolog was demonstrated in male flowers resulting exclusive expression of AP2 and PI that controlled stamen and carpel formation in the generative phase. Alterations of male and female floral meristem differentiation were demonstrated under photoperiodic and hormonal condition changes by applying in vitro treatments.

## INTRODUCTION

*Ambrosia artemisiifolia* L. is a monoecious, anemophilous plant from the *Asteraceae* family. It produces separate male and female flower heads in the same individual, being cosexual (*Lloyd, 1984*). This gender system shows 5% occurrence in sexually reproducing flowering plants mainly in gymnosperm pines and plants with catkins indicating an ancestral angiosperm state (*Renner & Ricklefs, 1995*). The numerous tiny male disc flowers are about three mm of diameter and grow in a terminal spike, subtended by joined bracts. The female flowers are one or two flowered and are inconspicuously situated below the male ones, in the leaf axils (*Bassett & Crompton, 1975*). Sepals are reduced and highly modified into pappus bristles. Petals form a single tubular flower, short anthers and long stigma (Fig. 1). One single plant alone may produce millions of pollen grains that is increasingly important from an allergological point of view in regions of Central and Eastern Europe. The distribution of common ragweed covers the area at medium latitude characterized by continental climate started its expansion from Hungary and meaning a current problem from both agricultural and public health aspects (*Makra et al., 2004*).

Since in the last 20–25 years the number of ragweed pollen has dramatically increased, it would be important to get a global attention and find the optimal control against the rapid spread of the plant caused by extraordinary plant-adaptability, large proportion of derelict land areas, the absence of specific pest insects and diseases, and not least the development of high-level resistance against herbicides. Considering the invasion migratory pathways, there is high risk for introduction of herbicide resistant genotypes from Eastern to Western Europe. Genomics based tools therefore play an increasing role in weed research (*Horvath, 2010*). Understanding the genetic control of sex determination during the plant life cycle may contribute to find an ecologically safer strategy in the common ragweed control.

Flower development is a complex and accurately coordinated biological and morphological process consisting spatial regulation of a considerable number of organ-specific genes during the life cycle of higher plants (*Taiz et al., 2015*). A large number of genes revealing floral meristem and structure formation were studied extensively using the model organism, *Arabidopsis thaliana* (*Komeda, 2004*), which is a monoecious long-day plant with unisexual flowers. Investigated genes discussed in this study were chosen based on the genetic knowledge of *Arabidopsis* flowering (*Chandler, 2011*; *Irish, 2010*). Based on *Arabidopsis* transcriptome analysis the majority of expressed transcripts were found in the reproductive part of flowers instead of the perianth, reflecting the more complex anatomy of tissue and cell types within stamens and carpels and major developmental events such as ovule and pollen formation (*Chandler, 2011*).

Functionally important genes (and their homologs) known to be involved in carpel and stamen formation are characteristic to the floral reproductive structures in *Arabidopsis thaliana*. Most likely MADS-domain transcription factors (TFs) control gene expression and hence the identity of floral organs and patterning during development. The transcription factor ABORTED MICROSPORES (AMS) (*Lou et al., 2014*; *Xu et al., 2010*) and genes such as LESS ADHERENT POLLEN3 (LAP3) (*Dobritsa et al., 2009*), LESS ADHESIVE

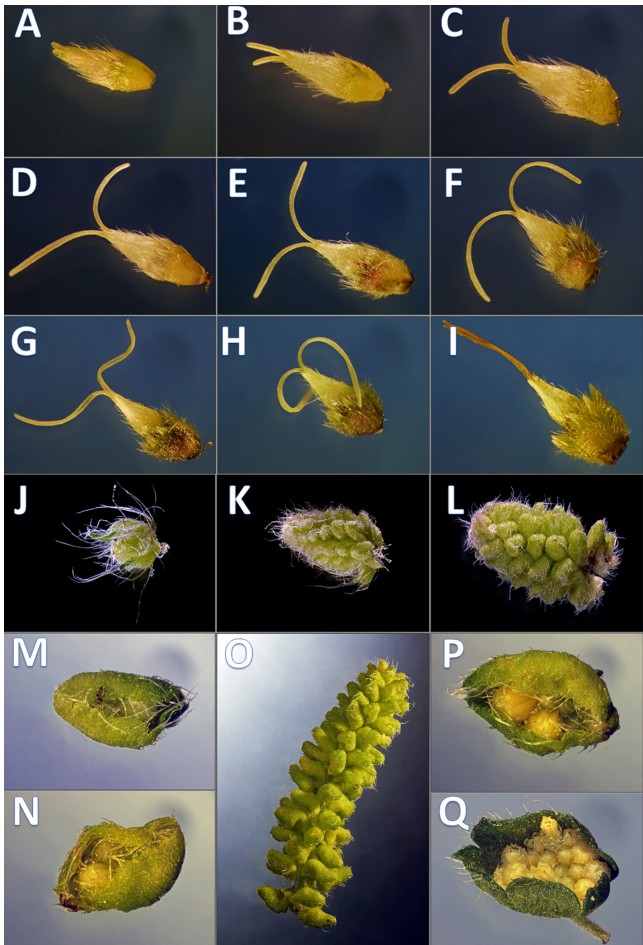

**Figure 1** **Developmental phenophases of female and male flowers used for RNA seq.** Symbols: Female flowers (F) and male flowers (M). For RNA extraction for *A. artemisiifolia* reference sequence the pool of F (A–I) and M (J–Q) of wild growing samples were performed, separately. In order to obtain pistillate specific transcriptomes during floral organ formation, RNA extraction were performed from in vitro samples from F (A–C) and F (G–I), wild growing samples from M (J–L) and M (P and Q) such as early and late developmental stages. Photo: Csaba Pintér.

POLLEN5 and LESS ADHESIVE POLLEN6 (LAP5, LAP6) (*Dobritsa et al., 2010*) encode anther-specific proteins required for pollen exine development, so called male candidate genes (*Rocheta et al., 2014*). MALE SPECIFIC EXPRESSION 1 (MSE1), is specifically expressed in male flowers. It is essential to early anther development (*Murase et al., 2017*). Expression of POLYGALACTURONASES (PG1 and PG2) and PECTIN METHYLESTERASE (PME) are characteristic in the tissues of mature pollen grains after microspore mitosis (*Futamura et al., 2000*). Female candidate genes such as filamentous floral proteins AXIAL REGULATOR YABBY1and YABBY4 (YAB1, YAB4) (*Meister, Kotow & Gasser, 2002*; *Siegfried et al., 1999*; *Villanueva et al., 1999*) and transcription factor AINTEGUMENTA (ANT) (*Elliott et al., 1996*; *Klucher et al., 1996*; *Long & Barton, 1998*) are relevant in pollen recognition, ovule formation, and embryo genesis identifying pistillate tissues. STIGMA SPECIFIC1 (STIG1) protein was found also as female

**Table 1 Function and expression values of male candidate genes in male (M), female (F) and leaf (L) libraries are shown in bold to validate floral transcriptomes.**

| Gene homolog | Encoded protein | Function (UniProt description) | Reference | Normalized expression | | |
|---|---|---|---|---|---|---|
| | | | | M | L | F |
| Male candidate genes | | | | | | |
| AMS | Transcription factor EN 48 | Regulates male fertility and pollen differentiation | Xu et al. (2010), Lou et al. (2014) | **8.91** | 0.44 | 0.52 |
| LAP3 | Protein strictosidine synthase-like 13 | Required for proper exine formation during pollen development | Dobritsa et al. (2009) | **9.61** | 0.21 | 0 |
| LAP5 | Type III polyketide synthase B | Required for pollen development and biosynthesis of pollen fatty acids | Dobritsa et al. (2010), Rocheta et al. (2014) | **6.65** | 0 | 0 |
| LAP6 | Type III polyketide synthase A | | | **8.39** | 0 | 0 |
| STIG1 | Stigma-specific protein | Involved in the temporal regulation of the exudate secretion onto the stigma | Rocheta et al. (2014) | **3.12** | 0 | 0 |
| PG1 | Polygalacturonase 1 beta-like protein 2 | Involved in cell size determination. | Futamura et al. (2000) | **3.42** | 0 | 0 |
| PG2 | Polygalacturonase 1 beta-like protein 3 | Involved in cell size determination. May serve as a chaperone for expansions through the secretory pathway | Futamura et al. (2000) | **4.71** | 0.43 | 0 |
| PME | Pectinesterase | Plays an important role in growth of pollen tubes. Involved in anther development and play role in tapetum and pollen development | Futamura et al. (2000) | **6.33** | 0.42 | 0 |
| MSE1 | Putative duplicated homeodomain-like superfamily protein | Exhibits tight linkage with the Y chromosome, specific expression in early anther development and loss of function on the X chromosome | Murase et al. (2017) | **7.12** | 0.41 | 0 |

**Note:**
STIG1 were found as male candidate instead of female as described in other species.

candidate in *Quercus suber* described by *Rocheta et al. (2014)* (Functional descriptions are summarized in Tables 1 and 2).

The dynamic process of flowering in *Arabidopsis thaliana* is induced from different pathways through the integration of environmental (photoperiodism and vernalization) and endogenous (hormonal regulation) signals. These signals trigger a complex system of interacting genes and proteins with repressor and activator roles influencing one another. *Corbesier et al. (2007)* identified a floral inducer FLOWERING LOCUS T (FT) playing an integral role between the different pathways (so called mobile "floral integrator"), that determine the floral development.

In angiosperms the circadian clock regulates floral induction in response to seasonal changes. In most cases the day-length as the main factor of photoperiodism may affect the flowering in plants. For example *Cannabis* sp., *Gossypium* sp., *Xanthium* sp. respond to short day signal, however, *Arabidopsis thaliana*, *Avena* sp., *Lolium* sp., *Trifolium* sp. respond to long day signals. In photoperiodic sensing there is a critical role of the CO gene.

**Table 2 Function and expression values of female candidate genes in male (M), female (F) and leaf (L) libraries are shown in bold to validate floral transcriptomes.**

| Gene homolog | Encoded protein | Function (UniProt description) | Reference | Normalized expression | | |
|---|---|---|---|---|---|---|
| | | | | M | L | F |
| Female candidate genes | | | | | | |
| YAB1 | Axial regulator YABBY 1 | In gynoecium, expressed in the abaxial cell layers differentiating into the valves meristem development | Siegfried et al. (1999) | 1.78 | 0.51 | **6.51** |
| YAB4 | Protein INNER NO OUTER | Essential for the formation and the abaxial-adaxial asymmetric growth of the ovule outer integument | Villanueva et al. (1999), Meister, Kotow & Gasser (2002) | 2.61 | 2.73 | **7.05** |
| ANT | AP2-like ethylene-responsive transcription factor ANT | Transcription activator. Required for the initiation and growth of ovules integumenta, and for the development of female gametophyte | Klucher et al. (1996), Elliott et al. (1996), Long & Barton (1998) | 2.04 | 0.35 | **7.91** |
| SUP* | Transcriptional regulator SUPERMAN | Transcriptional regulator considered as cadastral protein that acts indirectly to prevent the B class homeotic proteins APETALA3 and perhaps PISTILLATA from acting in the gynoecial whorl | Meister, Kotow & Gasser (2002) | 0 | 0 | **3.4** |
| TCP12* | Protein BRANCHED 2 | Transcription factor that prevents axillary bud outgrowth | Busch et al. (2011), Zhu, Sae-Seaw & Wang (2013) | 0 | 0 | **3.45** |
| MYB61* | Transcription factor MYB61 | Transcription factor that coordinates a small network of downstream target genes required for several aspects of plant growth and development | Arsovski et al. (2009), Liang et al. (2005) | 0.96 | 0.43 | **4.16** |
| MYB5* | Transcription repressor MYB5 | It is expressed in trichomes, seed coat and siliques | Romano et al. (2012), Yang et al. (2012) | 0.88 | 0.14 | **5.37** |

**Note:**
Genes marked with * were selected based on our filtering by using SQL Server Manager Studio (see "Materials and Methods").

Its gene product the CONSTANS protein is stabilized by light and initiate a complex interaction with inducers and repressors and binds to the FT promoter. Such a way may upregulate the floral inducer FTmRNA during circadian clock influenced by day-length (Putterill et al., 1995; Samach et al., 2000; Song, Ito & Imaizumi, 2010; Wigge et al., 2005). To induce flowering during long-day lighting conditions the CO protein activate the FT-mRNA in long-day plants; however, in the same conditions there is a repression process in short-day plants regulated by a light-stabilized CO ortholog Hd1, that repress the FT ortholog HEADING DATE 3 (Hd3a) (Hayama et al., 2003). With the illumination of far red and blue (late afternoon lightening conditions, daylight >700 nm) translation and stability of CO is regulated by PHYTOCHROME A (PHYA) and CRYPTOCHROME (CRY) repressing the CO degradation by CONSTITUTIVE PHOTOMORPHOGENIC 1 (COP1) and SUPPRESSOR OF PHYA proteins (SPA) (Jang et al., 2008; Laubinger et al., 2006; Yu et al., 2008). PHYTOCHROME B (PHYB) facilitates degradation of

CO with light conditions in the morning (daylight 640–680 nm) and delays flowering (*Song, Ito & Imaizumi, 2013*).

FLOWERING LOCUS C (FLC) is an epigenetic suppressor to inhibit floral induction in the vernalization pathway during the autumn period (*Amasino, 2010*) so called floral repressor after a cold treatment. In this period the prolonged cold exposure (10 to −1 °C) is required for the plant meristem to be competent to respond to floral induction conditions (*Simpson & Dean, 2002*). This is important to prevent flower development during the winter months when low temperatures may cause serious damage in floral tissue and most pollinators are absent. The FLC mRNA high level is maintained by the FRIGIDA (FRI) allele (*Gazzani et al., 2003*; *Michaels & Amasino, 2001*) downregulating the floral activators SOC1, FLOWERING LOCUS D (FD) and FT. It may bind directly to the promoters of SOC1 and FD and to a region in the first intron of FT, inhibiting the transcriptional activation of these genes and subsequent floral transition (*Helliwell et al., 2006*; *Searle et al., 2006*). VERNALIZATION INSENSITIVE 3 (VIN3) plays a central role in vernalization by mediating the initial transcriptional repression of the homeotic gene FLC (*Sung & Amasino, 2004*). However, due to its transient expression, it cannot maintain repression of FLC, which is then maintained by Polycomb Group complexes containing VRN1 and VRN2 throughout development. Required to deacetylate histones on the FLC promoter. Upon receiving the above described signals, under appropriate conditions, floral meristem identity genes (FMIGs) and the homeotic floral organ identity genes (FOIGs) are regulated by additional pathways. The upregulation of LEAFY (LFY) and APETALA1 (AP1) (FMIGs) evocate a cascade that regulates FOIGs expression (*Mandel & Yanofsky, 1995*; *Weigel & Nilsson, 1995*) toward floral whorl formation. LFY interacts with a number of homeotic genes to control their expression and therefore it is directly involved in the determination of floral architecture. Based on positive or negative regulation effect FMIGs can be divided into two distinct classes. The first class—such as LEAFY, APETALA1, and CAULIFLOWER—promotes flower meristem identity. The second class such as TERMINAL FLOWER (TFL1) has the opposite effect, and maintains the identity of inflorescence shoot meristems.

Hormonal regulation of generative development is highly dependent on the gibberellin (GA) signal and may alter in different species. GA involvement in floral initiation therefore is more complex. While GAs promote flowering in some LD and biennial species, their effect in other species are absent or may inhibit flowering of some perennials. In the case of SD plants the photoperiod flowering pathway is not decisive, therefore the GA pathway assumes a major role and becomes needful. GA INSENSITIVE DWARF (GID) proteins—encoded by GID1A, GID1B—are soluble GA receptors and bind with high affinity the biologically active GAs, which active state is catabolized by 2-beta-hydroxylation of GA 2-oxidases (GA2OXs). During flowering in response to GA, GIDs interact with specific DELLA proteins (e.g., GAI), known as repressors of GA-induced growth, and target them for degradation via proteosome (*Griffiths et al., 2006*).

During the processes of floral transition the cytokinins have important role as part of the hormonal regulation. It was demonstrated experimentally that the response to cytokinin treatment does not require FT, but activates its paralogue TWIN SISTER OF FT (TSF),

as well as FD, which encodes a partner protein of TSF, and the downstream gene SOC1 (*D'aloia et al., 2011*).

Floral organ identity determination is regulated by three classes of homeotic genes (FOIGs, encoding the A, B, and C functions). The ABC combinatory model (*Coen & Meyerowitz, 1991*) predicts that these classes act alone or combinatorially to give rise to sepals, petals, stamens, and carpels. The activities of class A such as APETALA (AP1, AP2), class B such as APETALA3 (AP3) and PISTILLATA (PI), and class C such as AGAMOUS (AG) are active in two adjacent whorls where their individual and combined activities specify the fate of organ primordia. A alone (whorl 1) specifies sepals, C alone (whorl 4) specifies carpels, and the combined activities of AB (whorl 2) and BC (whorl 3) specify petals and stamens, respectively. The A and C activities are mutually antagonistic, such that A prevents the activity of C in the outer two whorls, and C prevents the activity of A in the inner two whorls. The C activity is also required for floral determinacy: if the C activity is absent, an indeterminate number of floral whorls develop (*Pelaz et al., 2000*). SEPALLATA (SEP1, SEP2, SEP3, SEP4) genes form an integral part of models that outline the molecular basis of floral organ determination and are hypothesized to act as co-factors with ABC floral homeotic genes in specifying different floral whorls.

Reproduction genetics of *Ambrosia artemisiifolia* has been received poorly attention. Several study focuses on the pollen and seed production, pollen allergy (*Bassett & Crompton, 1975*; *El Kelish et al., 2014*; *Singer et al., 2005*; *Wayne et al., 2002*), life-cycle and reproduction strategy (*Kazinczi et al., 2008*) in *Ambrosia* spp. However, genetic information regarding flowering regulation is still rare.

Next generation sequencing (NGS) is an advantaged technology to capture the diversity of differentially expressed (DE) transcripts in male and female *Ambrosia artemisiifolia* flowers. In the present work we report Illumina RNA-sequencing of two developmental stages of female flowers compared with male and leaf transcriptomic data. Diversity of DE transcripts between female and male inflorescence as well as leaves were analyzed comparing with previously reported transcript datasets (*Virág et al., 2016*). Differences between expression levels of the above described flowering pathways and two gene categories such as MIGs and FOIGs (ABC ortholog—MADS box genes) identified in each type of flowers are also discussed. This study reports firstly about the sex-specific floral development in the common ragweed and underlying genetic expression pattern. The genetic events of staminate and pistillate inflorescence formation were investigated through expression analysis in wild growing and in vitro cultivated plants. Focusing on the role of female flower during plant reproduction and spread dynamic we present NGS data of early and late developmental phases of pistillate flower. This work and related NGS project are deposited in the National Centre for Biotechnology Information (NCBI) Bioproject under the ID PRJNA335689.

# MATERIALS AND METHODS

## Plant materials and cultivation conditions

The RNA-seq experiments were based on common ragweed samples from four bulked individuals. Seeds were collected from Cserszegtomaj, Zala county, Hungary (GPS: 46.79528,

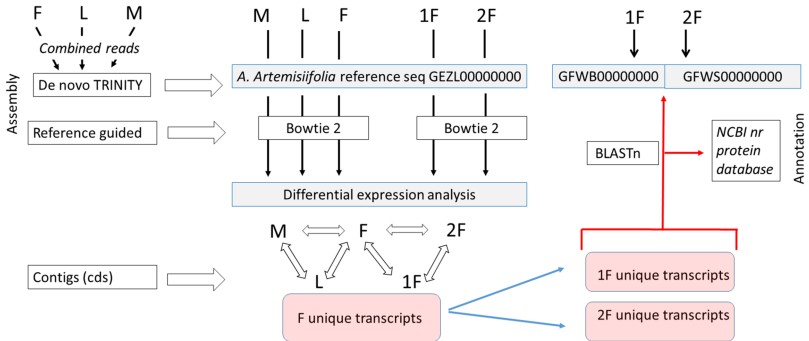

**Figure 2 Assembly strategy of *A. artemisiifolia* floral transcript datasets.** As a first step a de novo Trinity assembly of combined read set of male (M), female (F) and leaf (L) of *A. artemisiifolia* wild growing plants was performed to get a reference transcriptome that represent both vegetative and generative organ transcripts in different developmental stages. Separately two set of reads representing two classes of F development such as early (1F) and late (2F) stages were generated from sterile in vitro plants. In order to perform differential expression analysis of each five sample, cleaned read sets of M, L, F, 1F, and 2F were assembled by using bowtie2 reference guided algorithm aligning reads to the *A. artemisiifolia* reference transcriptome. Focusing on the female floral expression pattern in various developmental stages we determined the F unique transcripts collecting the whole female tissue specific contig set. After that 1F and 2F unique transcripts were clustered aligning them to the F contig set. In the second step Trinity de novo assembly were performed using 1F and 2F cleaned reads to get 1F and 2F specific contigs. In order to filter exactly the 1F and 2F specific transcripts, and characteristic transcription pattern to both developmental classes we aligned the 1F and 2F unique transcripts to 1F and 2F de novo Trinity contigs used Blastn algorithm. Thus filtered sequences were annotated using Blastx search and Blast2Go high throughput blast against the NCBI nr protein database. Transcript data sets of de novo Trinity assembly of combined F-L-M as *A. artemisiifolia* refseq and separately 1F and 2F were deposited in the NCBI TSA database with the accession number shown in the Figure.

17.26005) and were surface sterilized for 1 min in 70% (v/v) ethanol and washed with deep water for 23–30 min, followed by dipping in a 7% solution of calcium hypochlorite with TWEEN 20 for 20 min, and rinsing three times with sterile distilled water. The sterilized seeds were then germinated on MS medium (*Murashige & Skoog, 1962*) with 0.3 mg/l metatopolin hormon supplemented with 2% sucrose/litre and solidified with 0.8% phytoagar. The pH was adjusted to 5.8 prior to autoclaving at 120 °C for 20 min. All the cultures were maintained at 25 °C under a 16/8 light/dark photoperiodic regime with a light intensity of about 2,400–2,800 lux. Node segments (1.5–2 cm long) were excised from 21-day-old plants and transferred to above described MS medium.

Nine developmental stages of female flowers were collected from 12 in vitro cultivated individuals of common ragweed. The classification of the different phenological phases was based on visual observation (Fig. 1). For RNA-seq two categories of nine phenological phases (1F and 2F) were established and combined such as early 1F (Figs. 1A–1C) and late 2F (Figs. 1G–1I) developmental stages. Male flower buds were collected from genotypes used in F, M, L transcriptoms (*Virág et al., 2016*). For gene expression comparision we used 1M and 2M libraries as indicated in the Figs. 1J–1M, 1P, 1Q and 2M). In order to equal distribution, we determined the weight of the samples from each developmental stages (Table S1).

Along RT-qPCR analysis we used four additionally wild growing genotypes (collected in Keszthely, Zala county, Hungary, GPS: 46.7654716, 17.2479554).

## RNA-seq library construction and sequencing

Samples were frozen in liquid nitrogen immediately after collection. RNA extraction and on-column DNase digestion were performed from each categories using TaKaRa Plant RNA Extraction Kit (Takara Bio Inc, Kusatsu, Japan) following the manufacturer's instruction. RNA integrity and quantity were assessed on an Agilent 2100 Bioanalyzer (Agilent Technologies, Santa Clara, CA, USA).

Total RNA of 2.0 μg from 1F, 1M, and 2F, 2M samples were used as starting material for cDNA synthesis and library construction. Enrichment of mRNA, cDNA synthesis and library preparation for Illumina NextSeq paired-end sequencing were carried out using TruSeq™ RNA sample preparation kit (Low-Throughput protocol) using an oligo(dT)$_{18}$ as described by *Virág et al. (2016)*. The RNA-Seq was performed using Illumina NextSeq5000 system. Raw sequence data in FASTQ format were deposited in the NCBI sequence read archive (SRA) database under the accession SRR5965731 (1F) and SRR5965732 (2F). We used 1M and 2M for gene expression comparison in the case of investigated genes. 1M and 2M transcriptom datasets and their analysis will be published elsewhere.

Sequence reads of male inflorescence (M), leaves (L), and female (F) inflorescence containing all nine developmental stages were used from earlier reported and deposited data in NCBI SRA under the accession SRP08007 (*Virág et al., 2016*).

## Construction of *Ambrosia artemisiifolia* transcript datasets, identification of differentially expressed sequences and filtering pistillate specific transcripts

During preprocessing raw reads quality were examined using the FastQC quality control software (*Andrews, 2010*). Based on FastQC report, sequences found to be represented more than 0.1% of the total and low-quality bases (corresponding to a 0.1% sequencing error rate) were removed and trimmed using a self-developed application "GenoUtils" written in Visual Studio integrated development environment in C#. Conversion of .fastq to .fasta files was also performed using this in house application.

A sample-specific multiple assembly of cleaned reads from five libraries (M, L, F, 1F, and 2F) were performed. De novo assembly of each sample type were performed by using Trinity (*Haas et al., 2013*). For reference guided alignments Bowtie2 short read aligner (*Langmead et al., 2009*) was applied. The assembly strategy was performed in a way with taking into account the refinement of pistillate tissue specific sequences in both developmental classes as the final outcomes. Assembly strategy is summarized in the flow chart (Fig. 2). In detaile. First, a combined read set was assembled from the three sample libraries (M, F, and L) to generate a "reference" de novo transcriptome assembly that was deposited in the NCBI TSA database under the accession GEZL00000000 and reported in our prior study (*Virág et al., 2016*). During this processes, reads with certain length of overlap were combined to form contigs (kmer size, $K = 25$). This combined dataset

was used as reference shotgun assembly in the further alignments. In order to perform sample-specific expression analysis, we screened the DE sequences by aligning the original sample reads to the reference followed by abundance estimation using RSEM (*Li & Dewey, 2011*). The resulting DE transcripts were further clastered according to their expression patterns by applying Microsoft SQL Server Management Studio. Protein coding regions were extracted from the reference assembly using TransDecoder and further characterized according to likely functions based on sequence homology or domain content using BLAST+ (*Altschul et al., 1990*). In this way coding sequences (CDS) as unique transcripts of each five samples were screened and compared. Separately, de novo assemblies of 1F and 2F libraries were performed using Trinity and deposited in the NCBI TSA database under the accession GFWB00000000 (1F) and GFWS00000000 (2F). Using these reference contig sets 1F and 2F unique transcripts were realigned applying BLASTn with E value less than $10^{-5}$. The resulting narrowed and specified sequences were annotated using NCBI nr protein database.

## Selection of genes for floral transcriptome characterization

Genes representing different floral regulatory pathways—stimulated by environmental and endogenous signals—were selected for characterize genetic events during male and female floral development. Architecture formation were analized by determining ABC(E) genes expression pattern at generative growth (differences in FOIGs). Genes that enable and promote floral meristem initiation during vegetative growth (FMIGs) such as photoperiodic, vernalization, GA pathway, and flower development genes (Figs. 3 and 4) were also investigated in both wild growing and in vitro samples. Normalized expression of selected genes were performed digitally and expression values were compared.

## Digital gene expression analysis

The obtained reads from each library were mapped to the selected gene CDSs by using Bowtie2. The mapped reads were used to estimate the transcriptome level by the reads per kilobase per million mapped reads (RPKM) method (*Mortazavi et al., 2008*):

$$RPKM = \frac{CDS\ read\ count\ \times\ 10^9}{CDS\ length\ \times\ total\ mapped\ read\ count}$$

In this equation CDS represent the CDS of the investigated gene. Read count means the mapped reads from the total. CDS length is the nucleotid sequence length of coding region of the investigated gene and total mapped reads are the total raw reads of given cleaned library. For automatic calculation of RPKM values of genes a self developed pipeline as modul of GenoUtils were applied.

## Gene expression analysis with RT-qPCR amplification

Reverse transcription was performed starting from one μg total RNA using Maxima H Minus First Strand cDNA Synthesis Kit with dsDNase (Thermo Scientific, Waltham, MA, USA) according to the manufacturer's protocol, using an oligo(dT)$_{18}$ and random hexamer primers (Thermo Scientific, Waltham, MA, USA), final volume was 20 μl. cDNA (one μl)

| | Pathway | Gene | F | L | M |
|---|---|---|---|---|---|
| **Floral meristem initiation (vegetative growth)** | *Gibberellin* | GID1A | 83,32 | 353,76 | 160,19 |
| | | GID1B | 15,85 | 349,98 | 31,58 |
| | | GID2 | 40,18 | 187,02 | 73,88 |
| | | GAI | 36,01 | 540,92 | 73,89 |
| | | Ga2ox8 | 22,63 | 25,41 | 8,01 |
| | | Ga2ox1 | 8,44 | 86,51 | 39,89 |
| | | Ga2ox3 | 0,44 | 28,07 | 16,49 |
| | | Ga3ox1 | 0 | 67,16 | 9,65 |
| | | Cyp450 | 108,23 | 995,33 | 170,72 |
| | *Photoperiodic* | PHYB | 22,10 | 150,76 | 53,89 |
| | | PHYA | 2,36 | 5,76 | 8,03 |
| | | CRY1 | 14,37 | 533,24 | 38,92 |
| | | COP1 | 8,16 | 236,62 | 33,68 |
| | | CO | 9,52 | 2168,15 | 98,54 |
| | | SPA | 2,20 | 68,42 | 5,11 |
| | *Vernalization* | CSTF77 | 5,48 | 84,59 | 19,67 |
| | | CSTF64 | 3,72 | 81,72 | 17,07 |
| | | VIP6 | 5,56 | 94,84 | 22,94 |
| | | VIP5 | 11,96 | 181,12 | 41,91 |
| | | VIP4 | 13,34 | 189,45 | 61,25 |
| | | VIP3 | 41,96 | 98,16 | 61,81 |
| | | VIN3 | 0,00 | 313,45 | 10,81 |
| | | VRN1 | 9,09 | 41,66 | 27,08 |
| | *Flower development* | EMBF2 | 18,25 | 80,38 | 47,20 |
| | | CAL | 114,80 | 0,00 | 1,94 |
| | | SOC1 | 0,00 | 0,00 | 0,00 |
| | | FT | 0,00 | 1487,94 | 56,17 |
| | | FD | 0,00 | 10,40 | 16,05 |
| | | LMI2 | 6,75 | 7,12 | 41,52 |
| | | FUL | 6,17 | 103,31 | 75,64 |
| | | SVP | 12,51 | 31,71 | 7,55 |
| | | TFL1 | 0,00 | 1461,09 | 56,95 |
| | | SPL3 | 122,02 | 0,00 | 293,90 |
| | | SPL1 | 80,85 | 29,85 | 96,28 |
| **Floral architecture formation (flowering)** | **A** | AP2 | 0,00 | 110,81 | 1,98 |
| | **A** | AP1 | 162,27 | 1,93 | 6,11 |
| **ABC** | **B** | PI | 0,00 | 0,00 | 385,99 |
| | **B** | AP3 3 | 20,34 | 7,84 | 150,85 |
| | **B** | AP3 2 | 0,00 | 0,00 | 187,24 |
| | **B** | AP3 1 | 5,31 | 0,00 | 80,51 |
| | **C** | AG | 45,64 | 1,10 | 187,59 |
| | **E** | SEP4 | 111,34 | 1,10 | 43,86 |
| | **E** | SEP3 2 | 175,89 | 98,35 | 271,40 |
| | **E** | SEP3 1 | 78,92 | 7,52 | 59,01 |
| | **E** | SEP2 2 | 84,95 | 98,55 | 66,94 |
| | **E** | SEP2 | 105,90 | 1,90 | 32,51 |
| | **E** | SEP1 | 30,22 | 2,50 | 72,73 |

**Figure 3 Investigated genes representing different floral regulatory pathways in F, M, L libraries.** These genes were selected for characterize genetic events during male and female floral architecture formation and floral meristem initiation during vegetative growth such as photoperiodic, vernalization, and meristem identity genes. Normalized expression of selected genes were performed digitally and expression values were compared with each libraries F, M, L with HiSeq2000 technology.

| | Pathway | Gene | 1F | 2F | 1M | 2M |
|---|---|---|---|---|---|---|
| | **Gibberellin** | *GID1A* | 718,72 | 818,45 | 371,26 | 323,73 |
| | | *GID1B* | 60,25 | 188,07 | 77,34 | 64,89 |
| | | *GID2* | 520,22 | 462,20 | 228,17 | 236,33 |
| | | *GAI* | 306,10 | 295,90 | 407,41 | 420,34 |
| | | *Ga2ox8* | 207,07 | 220,68 | 37,58 | 33,71 |
| | | *Ga2ox1* | 83,26 | 48,55 | 147,25 | 157,31 |
| | | *Ga2ox3* | 18,87 | 15,10 | 41,42 | 69,05 |
| | | *Ga3ox1* | 0 | 0 | 162,01 | 177,76 |
| | | *Cyp450* | 139,44 | 270,7 | 1252,2 | 3499,78 |
| | **Photoperiodic** | *PHYB* | 298,71 | 244,20 | 224,23 | 209,08 |
| | | *PHYA* | 57,85 | 41,89 | 77,83 | 80,84 |
| | | *CRY1* | 230,51 | 424,48 | 95,38 | 102,69 |
| | | *COP1* | 74,27 | 64,88 | 102,09 | 96,59 |
| | | *CO* | 113,99 | 118,43 | 443,84 | 451,84 |
| | | *SPA* | 20,27 | 15,92 | 20,98 | 19,29 |
| | **Vernalization** | *CSTF77* | 126,07 | 114,60 | 140,87 | 130,34 |
| | | *CSTF64* | 91,56 | 67,24 | 55,48 | 55,49 |
| | | *VIP6* | 169,16 | 168,28 | 106,73 | 109,52 |
| | | *VIP5* | 153,24 | 177,16 | 144,57 | 151,45 |
| | | *VIP4* | 426,24 | 492,56 | 388,79 | 392,59 |
| | | *VIP3* | 218,86 | 156,07 | 137,23 | 128,29 |
| | | *VIN3* | 8,61 | 14,47 | 48,01 | 49,08 |
| | | *VRN1* | 177,92 | 273,02 | 100,43 | 103,99 |
| | **Flower development** | *EMBF2* | 143,32 | 128,64 | 114,59 | 107,31 |
| | | *CAL* | 2076,87 | 3059,01 | 15,11 | 13,49 |
| | | *SOC1* | 44,62 | 19,54 | 0 | 0 |
| | | *FT* | 7,21 | 0,00 | 295,35 | 258,03 |
| | | *FD* | 12,45 | 10,82 | 20,15 | 14,16 |
| | | *LMI2* | 73,47 | 29,53 | 188,22 | 208,34 |
| | | *FUL* | 81,42 | 70,65 | 316,24 | 298,81 |
| | | *SVP* | 56,48 | 49,01 | 24,26 | 15,75 |
| | | *TFL1* | 7,69 | 0,00 | 286,143 | 269,34 |
| | | *SPL3* | 364,35 | 602,39 | 635,25 | 686,31 |
| | | *SPL1* | 6,64 | 17,96 | 499,06 | 513,67 |
| | **A** | *AP2* | 0,00 | 0,00 | 4,97 | 5,59 |
| | **A** | *AP1* | 2435,41 | 3532,78 | 11,15 | 10,49 |
| | **B** | *PI* | 4,29 | 3,19 | 2370,99 | 2571,13 |
| | **B** | *AP3 3* | 229,98 | 247,66 | 994,37 | 1106,86 |
| | **B** | *AP3 2* | 163,94 | 223,67 | 1724,01 | 1757,56 |
| | **B** | *AP3 1* | 285,63 | 293,35 | 263,7 | 238,34 |
| | **C** | *AG* | 869,62 | 738,51 | 967,73 | 919,1 |
| | **E** | *SEP4* | 1285,10 | 1166,06 | 209,46 | 237,09 |
| | **E** | *SEP3 2* | 3073,76 | 1974,46 | 1031,59 | 1097,51 |
| | **E** | *SEP3 1* | 661,21 | 499,63 | 215,78 | 217,48 |
| | **E** | *SEP2 2* | 1229,70 | 1530,87 | 278,9 | 276,62 |
| | **E** | *SEP2* | 2060,41 | 1628,81 | 222,62 | 238,75 |
| | **E** | *SEP1* | 732,25 | 534,94 | 915,63 | 964,9 |

*(Left margin labels: "Floral meristem initiation (vegetative growth)" spanning the Gibberellin, Photoperiodic, Vernalization, and Flower development pathways; "Floral architecture formation (flowering) ABC" spanning the A–E rows.)*

**Figure 4 Investigated genes representing different floral regulatory pathways in 1F, 2F, and 1M 2M libraries.** The normalized expression values of flower development genes has also been studied in libraries 1F, 2F, 1M, 2M with NextSeq 500 technology derived from in vitro and wild growing plant materials.

was used for real-time PCR amplification on a Bio-Rad CFX96 System. qPCR analysis (and efficiency) was performed with one µl of cDNA on a Bio-Rad CFX96 System using Xceed qPCR SG Mix (Institute of Applied Biotechnologies, Praha, Czech Republic). The relative gene expression was calculated with ΔΔCt method using Bio-Rad CFX Manager™ Software v3.1. Primer efficiency was analyzed with CFX Manager™ Software v3.1 (Bio-Rad, Hercules, CA, USA). PCR was performed as follows: an initial activation of the polymerase enzyme at 95 °C for 2 min was followed by 46 cycles at 95 °C 5 s, 59 °C for 30 s, ended with a 5 s melt analisys ranging between 65 and 95 °C with 0.5 increments. Primers (Table S2) were designed with Primer3 (*Koressaar & Remm, 2007*; *Untergrasser et al., 2012*) based on in silico sequence prediction of CDS.

Gene expression analysis was established based on three technical and biological replicates, and normalized with the reference gene glyceraldehyde-3-phosphate dehydrogenase

(*GAPDH*) (*Hodgins et al., 2013*). In a pilot experiment *GAPDH* was selected as reference gene. In this investigation based on the literature CDS of 15 housekeeping genes were determined. Subsequently the relative expression levels were determined in three sample types (L, M, F) were six reference genes showed the same level, respectively. Among the equally expressed genes three reference genes (*GAPDH, TUA, TUB*) were selected for RPKM value calculation (Table S3) and PCR experiments. *GAPDH* was the most characteristic along these investigations. Based on RPKM values *GAPDH* expression referd on equivalent level order of magnitude level in each liberies ranging from 2,480 to 4,914.

### Annotation of DE transcripts

Whereas *Ambrosia artemisiifolia* floral transcriptome was a newly targeted transcriptome the annotation procedure was performed as described by *Haas et al. (2013)*. Potential coding regions within reconstructed transcripts that were insufficiently represented by detectable homologies to known proteins were predicted based on metrics tied to sequence composition by applying TransDecoder include with Trinity. Running this application on the Trinity-reconstructed transcripts the candidate protein-coding regions could be identified based on nucleotide composition, open reading frame (ORF) length, and (optional) Pfam domain content. To predict their functions the latest non-redundant protein database 10/03/2016 (ftp://ftp.ncbi.nlm.nih.gov/blast/db/FASTA/nr.gz) were used.

For further annotation of unigenes using various bioinformatics approaches, the unigenes were firstly searched against the Nr database and the Swiss-Prot protein database using local BLASTx (*Gish & States, 1993*) with $E$ value cutoff of $10^{-5}$. With Nr annotation, Blast2GO (*Götz et al., 2008*) was used to get gene ontology (GO) annotation according to molecular function, biological process and cellular component ontologies (http://www.geneontology.org).

## RESULTS

In this study we present an RNA-seq approach to get closer to the mechanism of floral development in *Ambrosia artemisiifolia* using Illumina sequencing. In order to investigate the expression distribution of genes taking part in floral morphogenesis, we categorized transcript datasets of female and male flowers including whole investigated phenophases (F, M), leaves (L) and female and male flowers in early and late developmental stages (1F, 2F, 1M, 2M). For normalized F, M, and L transcript libraries, we used the SRA dataset reported in our prior study (*Virág et al., 2016*) provided the first step in the common ragweed floral genomics. For 1F, 2F, 1M, 2M libraries early and late categories of phenophases were collected and sequenced de novo from in vitro and wild growing plants.

### RNA sequencing and assembly of *Ambrosia artemisiifolia* floral libraries

The M, F, and L libraries were obtained through HiSeq2000 platform. Library construction and assembly was described detailed in our prior study (*Virág et al., 2016*). In brief, after cleaning we obtained 18,472,374 (M); 15,290,200 (F) and 17,435,976 (L) from total raw reads of 24,110,256; 23,264,636 and 24,330,693 (2 × 100 bp). Assembly of combined read
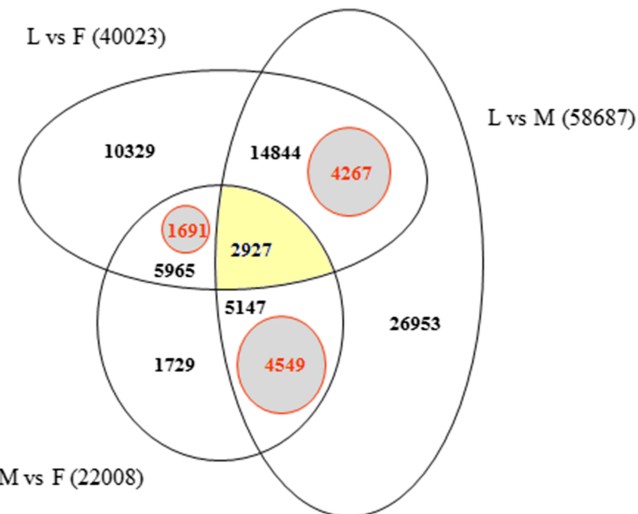

**Figure 5 Venn diagram indicating the number of exclusive and shared transcripts in wild growing male (M), female (F) flowers and leaf (L) samples.** The datasets were extracted from Trinotate heat map and visualized with FunRich. The comparison and distribution of transcripts were specified according the expressed sequences in groups of leaf vs. female, male vs. female and leaf vs. male transcriptomes of *A. artemisiifolia* reference GEZL00000000. The SQL queries for organ specific unique transcripts resulted in 4,549 (unique M), 1,691 (unique F) and 4,267 (unique L).

sets resulted in 229,116 transcripts and 162,494 trinity genes (unigenes) and used as a reference (Table S4).

A total of 39,664,366 (1F) and 37,127,852 (2F) 2 × 80 bp reads were generated by the Illumina NextSeq500 system. High quality (> Q20) bases were more than 91% in (paired-end) reads of both sample. Percentage of unresolved bases (Ns) and overrepresented sequences was observed to be minimal 0.197% and 0.0004% in average. In total of 92% were high quality sequences from raw data: numerically 36,491,216 (1F) and 34,157,623 (2F). These high quality, processed paired-end reads were used to assemble into contigs and transcripts. Their de novo assembly resulted in 109,452 (1F) and 97,239 (2F) contigs. In comparision the reference guided assembly led to 40% more contigs 147,457 (1F) and 141,58 (2F), respectively. However, N50 length showed a minor 5–11% dispersion between the ranges 626.9–697.5 in both methods. Sequencing statistics are summarized in Table S4.

### Selection of *Ambrosia artemisiifolia* unique and differentially expressed genes

In order to identify exclusively expressed transcripts in the sex-specific floral tissues we used Bowtie2 alignments of F, M, L, 1F and 2F libraries for the *Ambrosia* reference transcriptome GEZL00000000. In order to prove the 1F and 2F exclusive transcripts in female sample (F), contigs were realigned using BLASTn algorithm to the 1F and 2F de novo Trinity transcripts (GFWB00000000, GFWS00000000). Filtering method is summarized in Fig. 2. The transcript datasets were extracted from Trinotate heat map and postprocessed using Microsoft SQL Server Management Studio (Dataset S1). The queries for unique organ specific sequences resulted in 5,659 (M), 1,691 (F), and 4,267 (L).

**Table 3  Genes expressed exclusively in female transcriptomes (F, 1F and 2F).**

| Gene homolog | Encoding protein | RPKM value (F) HiSeq2000 | RPKM value (1F) NextSeq500 | RPKM value (2F) NextSeq500 | Function | Reference |
|---|---|---|---|---|---|---|
| *PBL9* | Serine/threonine protein kinase | 88.97 | 640.35 | 57.32 | Exhibits serine/threonine activity | *Hirayama & Oka (1992)*, *Ito et al. (1997)* |
| *SUP* | Transcriptional regulator SUPERMAN | 7.52 | 123.31 | 56.55 | Acts indirectly to prevent AP3 and perhaps PI proteins from acting in the gynoecial whorl | *Kazama et al. (2009)*, *Bowman et al. (1992)* |
| *ACA7* | Calcium-transporting ATPase 7, plasma membrane-type | 22.57 | 0 | 4.92 | Carbonic anhydrase that catalyses the reaction of reversible hydration of $CO_2$ | *Henry (1996)*, *Smith & Ferry (2000)* |
| *TCP12* | Transcription factor TCP12 | 12.6 | 239.69 | 134.51 | Prevents axillary bud outgrowth, delay early axillary bud development | *Busch et al. (2011)*, *Zhu, Sae-Seaw & Wang (2013)* |
| *MYB5* | Transcription repressor MYB5 | 19.01 | 122.98 | 75.4 | Involved in seed coat formation, trichome morphogenesis and mucilage secretion | *Romano et al. (2012)*, *Yang et al. (2012)*, *Li et al. (2009)*, *Liu, Jun & Dixon (2014)* |
| *WIP2* | Zinc finger protein WIP2 | 7.29 | 131.11 | 114.05 | Required for normal differentiation of the ovary transmitting tract cells | *Crawford & Yanofsky (2008)*, *Marsch-Martínez et al. (2014)* |
| *MYB61* | Transcription repressor MYB61 | 7.59 | 205.63 | 349.61 | Required for seed coat mucilage deposition | *Arsovski et al. (2009)*, *Liang et al. (2005)* |
| *LAC2* | Laccase-2 | 30.84 | 0 | 3.93 | LAC2 deficient lead to early flowering | *Tashian (1989)* |
| *BEE* | Transcription factor BEE1 | 5.29 | 0 | 59.63 | Positive regulator of brassinosteroid signaling (BR) | *Domagalska et al. (2007)* |

The number of exclusive and shared transcripts in wild growing male and female flowers and leaf are visualized in Venn diagram. For visualization we used FunRich (V3) software (*Pathan et al., 2015*) (Fig. 5). The number of annotable unique transcripts were 11,300 in M, F, and L libraries. Among which the most important organ specific genes are summarized in Tables 3 and 4 and their sequences are in Dataset S2. The presented genes were within the upper limit (>80%) of the total expression level value.

## Annotation and functional classification of *Ambrosia artemisiifolia* unigenes

Functional annotation of the transcriptome sequences and analysis of the annotation of three-specific RNA pools (M, F, L) were performed using BLASTx search (*Gish & States, 1993*) and Blast2GO (*Conesa et al., 2005*). This process resulted in 81.56% translated contigs in total. Based on GO annotation the cell part, organic cyclic compound binding and organic substance metabolic process were the most abundant GO slims in the categories of cellular component, molecular function, and biological process, respectively (Fig. S1). The high physiological activity of *Ambrosia* reproductive tissues was suspected from the high representation of primary metabolic process and cellular metabolic process groups within the biological process category.

**Table 4 Genes expressed exclusively in male transcriptomes (M, 1M, 2M).**

| Gene homolog | Encoding protein | RPKM value (M) HiSeq2000 | RPKM value (1M) NextSeq500 | RPKM value (2M) Nextseq500 | Function | Reference |
|---|---|---|---|---|---|---|
| *pcC13-62* | Desiccation-related protein | 1,126.37 | 4,335.81 | 5,839.66 | Expressed exclusively in the stylopodium | *Zha et al. (2013)* |
| *CYP450 86B1* | Cytochrome P450 | 906.19 | 4,304.38 | 3,499.78 | Effect the suberin and cutin biosynthesis | *Duan & Schuler (2005)*, *Compagnon et al. (2009)* |
| *GDSL2* | Lipase/Acylhydrolase superfamily protein | 804.54 | 2,230.36 | 1,603.26 | Expressed during petal differentiation | *Ji et al. (2017)* |
| *OAS* | 3-oxoacyl-[acyl-carrier-protein] synthase | 737.44 | 326.14 | 303.02 | Catalyses the condensation reaction of fatty acid synthesis | *Dayan & Duke (2014)*, *Hakozaki et al. (2008)* |
| *PMADS2* | PMADS 2 | 309.49 | 1,968.72 | 2,038.71 | Pi (Pistillata) homolog, predominantly expressed in petals and stamens | *Van Der Krol et al. (1993)* |
| *MYB80* | Transcription factor MYB80 | 170.82 | 440.74 | 381.95 | Play an essential role during anther development | *Higginson, Li & Parish (2003)*, *Phan et al. (2011)* |
| *MYB35* | Transcription factor MYB35 | 144.81 | 426.79 | 398.56 | Required for anther development and early tapetal function during microspore maturation | *Zhu et al. (2011)* |
| *TET8* | Tetraspanin 8 | 140.82 | 236.92 | 250.22 | Expression was observed during the pollen development and in mature pollen | *Reimann, Kost & Dettmer (2017)*, *Honys & Twell (2004)*, *Pina et al. (2005)*, *Boavida et al. (2013)* |
| *MYB44* | Transcription factor MYB44 | 99.18 | 137.58 | 135.46 | Represses the expression of protein phosphatases 2C in response to abscisic acid | *Jung et al. (2008)* |
| *NIP* | Aquaporin | 49.68 | 51.78 | 25.27 | Play role in pollen germination and pollination | *Di Giorgio et al. (2016)* |
| *SYN* | Syntaxin | 45.70 | 80.96 | 76.05 | Syntaxins mediate membrane fusion | *Sanderfoot et al. (2001)* |
| *MYB26* | Transcription factor MYB26 | 15.57 | 183.91 | 212.97 | Regulates lignified secondary cell wall thickening of the anther endotechium | *Mitsuda et al. (2006)*, *Yang et al. (2007)* |

Representation of the significant sub-ontologies of cellular component, molecular function, and biological process are summarized in Fig. S1. GO enrichment analysis were performed with the investigated floral genes wich resaults are summarized in Figs. S2–S4. Statistics and background information can be found in Datasets S4 and S5.

## Validation of the *Ambrosia artemisiifolia* floral transcriptome

The identification of differences between the male and female floral transcriptomes was based on in silico analysis. We performed a local blast alignment by using the determined 80 CDS fasta sequences (MK098047–MK098126) against male (M) and female (F) transcriptoms. No qualitative differences were found among the sequences of commonly expressed genes in two floral organs. The blast output files are in Dataset S3. The observed expression differences were validated investigating the transcript expression levels of homolog genes responsible for pollen and embryo formation in model organism

*Arabidopsis* thaliana. The reliability of male transcriptome was characterized by overexpression of the homologs of *AMS, LAP3, LAP5, LAP6, MSE1, PME, PG1, PG2*. The female transcriptome validity was justified by overexpression of *YAB1, YAB4*, and *ANT* transcription factors. Additionally, we introduced *SUPERMAN* (*SUP, MK098096, GO: 0003676*), *Transcription repressor MYB5* (*MYB5, MK098099, GO:0003677*), *Transcription repressor MYB61* (*MYB61, MK098101, GO:0003677*) and *Protein BRANCHED2* (*TCP12, MK098098, GO:0003700*) as female candidate genes based on our filtering by Microsoft SQL Server Management Studio. These regulators are involved in plant and floral tissue development, but their role during sex determination is still unclear. Expression differences and function of organ specific genes are summarized in Tables 1 and 2.

### RT-qPCR validation of tissue specific genes

For the RT-qPCR validation we selected the genes showing the largest differences (Tables 1 and 2) and most interesting from functional aspects. *TCP12, SUP*, and *LAP6* were selected to justify their exclusive expression in female and male flowers. *STIG1* were chosen according to its opposite expression in male flowers. Additionally we selected also *PI* (*MK098083, GO:0000977*) as ABC transcript exclusively present in staminate tissues.

### Gene expressions during floral meristem initiation (vegetative growth)

Functional and structural analysis of *Arabidopsis* thaliana gene homologs were investigated to elucidate floral regulation in the common ragweed. Expression pattern of photoperiodic, vernalization, GA, and flower development pathways comparing with floral architecture formation are summarized in Figs. 3 and 4.

### Gene expressions during floral architecture formation (generative growth)

The ABC(E) genes responsible for the formation of flower organs were determined on the basis of various compositae species (*Gerbera sp, Helianthus sp, Tagetes sp, Chrysanthemum sp*) found in public databases (NCBI, UNIProt). The CDSs and their expression patterns of the following 13 gene homologs were determined: *AP1* (*MK098082*), *AP2* (*MK098081*), *AP3 1* (*MK098086*), *AP3 2* (*MK098085*), *AP3 3* (*MK098084*), *PI, AG* (*MK098087*), *SEP1* (*MK098094*), *SEP2 1* (*MK098092*), *SEP2 2* (*MK098091*), *SEP3 1* (*MK098090*), *SEP3 2* (*MK098089*) *and SEP4* (*MK098088*) (Figs. 3 and 4). All of investigated floral organ genes except *AP2* (*GO:0003677*) have the same GO ID: GO:0000977.

### Transcription factors differentially expressed in F, M, and L libraries

Based on analysis of the different TF groups that are sex-specific and DE in each library the following factors were identified in our samples according to *Rocheta et al. (2014)*. Gene expression values of all investigated libraries and functional descriptions are summarized in Tables S5 and S6.

### Expression of female flower specific genes

In order to identify exclusively expressed genes in female flowers, the combined assembly of five (M, L, F, 1F, and 2F) non-normalized libraries were analyzed. In the analysis,

1,691 unique transcripts were found in the wild growing F library, representing the nine developmental stages. Transcripts longer than 700 bp were further investigated. After this filtering, we found 60 transcripts showing evaluable ORF, among which nine genes were well annotable using NCBI nr and Swiss-Prot databases. Based on this, the following protein coding and TFs were identified: *Protein kinase 1A* (*PBL9, MK098095, GO:0004674*), *SUP, Alpha carbonic anhydrase-7* (*ACA7, MK098097, GO:0004089*), *Laccase-2* (*LAC2, MK098102, GO:0005507*), *TCP12, MYB5, MIB61, Zinc finger protein WIP2* (*MK098100, GO:0003677*), and *Protein Brassinosteroid enhanced expression 1* (*BEE1, MK098103, GO:0006355*) (see Table 3). In order to find genes that might correspond to early and late flower development, exact CDSs of these genes were determined in silico and validated with sanger sequencing from in vitro plant materials. RPKM values were calculated in F, L, M, 1F, 2F, 1M, and 2M libraries using validated CDSs.

## Expression of male flower specific genes

The uniquely expressed transcript number in male tissues were 4,549 (Fig. 5) of which sequences longer than 700 bp were filtered out. In this way total CDSs of 41 genes were identified based on NCBI nt Blast and NCBI ORF finder databases. The 12 most characteristic male specific genes, RPKM values and functional properties are summarized in Table 4.

## DISCUSSION

Genetic control of floral morphogenesis is a well investigated field in the model plant *Arabidopsis* thaliana. Using *Arabidopsis* hermaphrodite flowers and their mutants some genetic models were established for structural development of floral organs in which a series of sequential steps were defined; for review, see *Krizek & Fletcher (2005)*. As the initial destination the floral meristem are regulated through the activation of FMIGs by flowering pathways. Subsequently, the floral meristem is patterned into the whorls of organ primordia through the activity of FOIGs. Then the FOIGs activate downstream effectors that specify the various tissues and cell types that constitute the different floral-organ types. Each of these steps are under strict genetic network control of positive and negative factors and regulatory elements interacting at various levels to regulate floral morphogenesis. Using NGS RNAseq, there is a great possibility to investigate expression profile of these pathways in non-model organisms like *Ambrosia artemisiifolia*, where the male and female floral organs grow as separate entities in the same individual plant. *Rocheta et al. (2014)* report on a comparative study revealed a transcript dataset to be involved in flower and plant development in the monoecious tree *Q. suber* with similar disjunct gender system where the male flowers form catkins and female inflorescence grow on young leaf axils separately. This floral transcriptome analysis revealed a group of genes expressed exclusively in each type of flower genders that may have a functional role in the sexually different floral organ development and sex specification (*Rocheta et al., 2014*).

## Validation of the *Ambrosia artemisiifolia* floral transcriptome

As expected, the investigated genes showed differential expression ratios with a meaningful results according that male tissues were characterized with an overexpression of genes (*AMS, LAP3, LAP5, LAP6, STIG1, PG1, PG2, PME, MSE1*) taking part in pollen structural

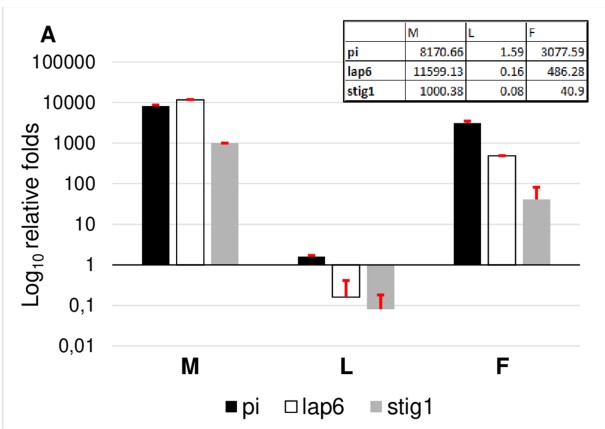

|  | M | L | F |
|---|---|---|---|
| pi | 8170.66 | 1.59 | 3077.59 |
| lap6 | 11599.13 | 0.16 | 486.28 |
| stig1 | 1000.38 | 0.08 | 40.9 |

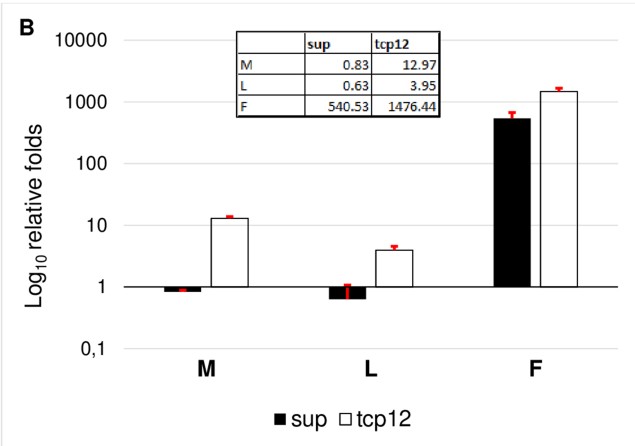

|  | sup | tcp12 |
|---|---|---|
| M | 0.83 | 12.97 |
| L | 0.63 | 3.95 |
| F | 540.53 | 1476.44 |

**Figure 6 Comparative analysis of male specific (A) and female specific (B) gene expression.** RT-qPCR analysis were determined as $\log_{10}$ relative fold. Tables inserted in the diagram show the real values of expression measured by RT-qPCR.

development, fatty acid and exine constituent biosynthesis during the anther and pollen differentiation. In contrast, expression of these genes was completely absent in the pistillate tissues in which ovule polarity, initiation, structural development and embryogenesis influencing genetic elements (*YAB* class TFs) showed about 67% higher presence than in staminate samples. The non-exclusive occurrence of these sequences in pistillate flowers may be explained by the contribution to setting up the polarity in stamen maybe through their action in supporting the development of the floral meristem or/and regulating margin growth (*Eshed et al., 2004*) that was observed also in all lateral organs (*Chen et al., 1999*; *Siegfried et al., 1999*). Additionally, the validation method was confirmed through the very low incidence of candidate sequences in leaf transcriptomes.

## RT-qPCR validation of tissue specific genes

To evaluate the results, genes were grouped according to their specificity. According to this, the female transcriptome was validated based on TCP12 and SUP expressional pattern. Female specificity of SUP and TCP12 was confirmed (Fig. 6B). The expression of

TCP12 was three times higher than of the SUP in female flowers confirming the in silico RPKM results. Male transcriptome was validated according the expression pattern of LAP6, STIG1, and PI. The non exclusive expression of PI was observed indicating the ABC function and B class activity in both flower types. Since in female flower the petal differentiation is vestigial and this whorl is united into a single tubular flower, therefore PI homolog transcript are less present than in male flowers where petals are united into short anthers with long stigma and stamen structures represents a large amount.In silico expession of STIG1 homologe in male flowers (Table 1) were confirmed by RT-qPCR, indicating the modified pistil origin in the pistillodium structure. Howerver this transcript showed a minimal expression also in female flowers (Fig. 6A). This fenomena reveals the common origin of these flower types which were separated during evolutuion.

## Gene expressions during floral meristem initiation (vegetative growth)

Plant development and architecture is regulated by meristems that initiate lateral organs on their flanks. Gene regulatory networks that control the transition of a vegetative shoot apical meristem into an inflorescence meristem (FMIGs), together with those necessary to specify floral meristem identity (FOIGs) have been explored in *Arabidopsis* thaliana and are highly complex and redundant. We have evaluated and report the most prominent cases in this study. Since the morfogenesis of the in vitro cultivated female inflorescence of common ragweed is more intensive than in wild-growing plants, therefore genetic regulation of the pistillate inflorescence can be more effectively investigated in these samples. The investigated pathways and related genes showed the following correlations. No interaction was found between *LFY, FRIGIDA (FRI)* and *FLC* transcription factors that determine the initiating steps during vegetative growth. Homologous transcript of this genes were not found in any of the investigated libraries, and were not expressed in the examined samples. This indicates, that the blossom phase has exceeded the initial signals in the bio-rythm of collected samples. The combined expression of *LFY* and *FLC* leads to the so-called *"late-flowering phenotype"* (Michaels & Amasino, 2001) absent in the 1F and 2F individuals confirming the complete lack of expression of these genes, that may be traced back to the altered hormonal conditions (cytokinin-supplemented culture medium) during in vitro cultivation.

Overexpression of **photoperiodic pathway** related transcripts of *PHYB* (MK098056, GO:0000155) in 1F, 2F, 1M and 2M libraries and *COP1* (MK098059, GO:0009785), *CO* (MK098060, GO:0005634) *and FT* (MK098073, GO:0008429) in the L library were found. Significantly higher expression of *PHYB* than *PHYA* (MK098057, GO:000155) were observed in all samples independently of the cultivation conditions indicating the red (morning) illumination affects more intensively the flowering processes in *Ambrosia artemisiifolia*. These results confirm the findings of *Deen, Swanton & Hunt (2001)* at genetic level, according which *Ambrosia artemisiifolia* belongs to the short day plants. The transcription of *FT* is activated by CO in leaf, and the FT protein moves to the shoot apex to induce flowering, therefore we found these transcripts overexpressed only in leaf tissues in wild-growing samples. Since *Ambrosia artemisiifolia* is a short day plant, higher expression of *COP1* in leaf may be explained by the function to repress CO

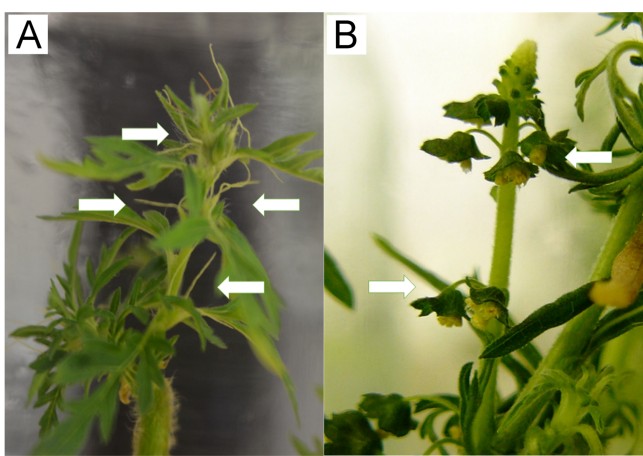

**Figure 7 In vitro cultivated *A. artemisiifolia* individuals.** During the life cycle of in vitro plant the female flowers were appeared earlier and in greater amounts (A). Appearence of male inflorescences were minimal (B) which is the opposite phenomena that may observed under wild-growing conditions. Arrows show the appearance of dense pistillate and spars staminate flowers in vitro.

degradation (*Song, Ito & Imaizumi, 2010*). Interestingly, FT-mRNA was only detected in early stage inflorescences (1F) suggesting that the initial flowering signal loses its regulatory function during female flower development, and TFs responsible for floral organ development come to the fore as control elements.

The **vernalization pathway** gene homolog *VRN1* (*MK098069, GO:0003677*), showed high values in all developmental stages (1F, 2F, 1M, 2M), which function is to inhibit the flowering repressor *FLC*, thus maintain generative morphogenesis of the flowering meristem. Expression of *VRN1* was increased during female flower maturation (*Sung & Amasino, 2004*). We found *SOC1* (*MK098072, GO:0000957*) homolog in 1F and 2F libraries. *SOC1* can integrate flowering signals from different pathways and interact with FT-mRNA to develop inflorescence. The *SOC1-FT* interaction, based on gene expression values, showed intensive regulation in the early stages of the common ragweed (1F).

*SOC1* associating with *AGL24* (*MK098093, GO:0000977*) mediates effect of GAs on flowering under short-day conditions, and regulates the expression of *LFY*, which links floral induction and floral development (*Liu et al., 2008*). The lack of *SOC1, LFY* (RPKM = 0) and overexpression of *FT* (RPKM = 1,487) in wild-growing samples indicated that flowering transition was regulated more intensively by the photoperiodic pathway in this samples. In contrast, hormonal pathway related genes (*SOC1, GID1A* (*MK098047, GO:0016787*), *GAI* (*MK098050, GO:0003712*), *Ga2ox8* (*MK098051, GO:0016491*), *CAL* (*MK098071, GO:0000977*), *AGL24*) were more expressed in in vitro plants. Therefore we concluded, that both cultivation condition reached the *AP1* expression that is the "first pass" of floral architecture formation, however, the vegetative pathway was altered under in vitro conditions. This phenomena has led to altered morphology and genesis of floral genders. The female inflorescences grow faster and in greater abundance in the case of in vitro individuals than in the wild-growing plants (see Fig. 7). Male inflorescences,

however, were barely appeared in vitro. Under wild-growing conditions male flowers dominate because of the intense lightening mediated signals.

The effective integration of **hormonal signaling** in plant vegetative growth is essential in which GAs and cytokinins act antagonistically in leaf formation and meristem maintenance (*Gan et al., 2007*). Cytokinins may act antagonistically with GA preventing *LFY* induction (*Li, Li & Smith, 2017*) that was observed in 1F and 2F samples in our flowering system. However, their integrative role in flower formation is still contested. We demonstrated the importance of cytokinins as part of hormonal regulation during *Ambrosia* floral transition (overexpression of *TSF* in L library), however, no sex-specificity was observed (low RPKM values in 1F, 2F, 1M, 2M libraries). We demonstrated also that cytokinin signaling may act positively on GA-pathway genes during flowering therefore GA pathway genes (*GID1A* and *GID2* (MK098049, GO:0005515)) were found overexpressed in female inflorescences.

## Gene expressions during floral architecture formation (generative growth)

The *AP2, AP3 2*, and *PI* genes were not or very low expressed in the female samples, therefore we considered them as male specific genes of common ragweed. *PI* with *AG* determine the stamen and *AP2* with *AP3* determine the petal formation that are male flower elements in the common ragweed (Fig. 8A). The expression of the *AP1* gene showed a significantly higher value in female flowers than in the male inflorescence, suggesting that the initiation of petals and stamens is stronger in the pistillate flowers. This overexpression may be traced back to the hormonal pathway domination and through *SOC1* during vegetative phase in female flowers (*Kaufmann et al., 2010*). In the early stage of development of female inflorescences (1F) a high expression rate were found for *SEP1, SEP2, SEP3*, and *SEP4*. Supposedly these genes are responsible for the development of petal and carpel primordias. Hovever, *SEP3 2* and *SEP2 2* showed high expression in the late phase (2F) that may be involved in the pistill prolongation (Fig. 8B).

## Transcription factors differentially expressed in F, M and L libraries

Transcription factors play important roles in plant development and flower morphogenesis responding to the environment. These factors interact specifically with sequences located in the *promoter regions* of the genes they regulate. TFs are classified in families according to the structure of their *DNA-binding domain*.

Based on RPKM values, homologs of *CONSTANS-LIKE* zink-finger TF family was found to be characteristic in both female and male samples (*COL4, MK098124, GO:0005634, COL5, MK098125, GO:0005634*). Upregulation of *COL4* was found in 1F and 2F libraries indicating a stronger interaction of female tissues with the circadian clock and light signals than in male flowers. *COL4* null mutants flowered early under short or long days. In contrast, *OsCOL4* activation-tagging mutants *(OsCOL4-D)* flowered late in either environment (*Lee et al., 2010*). Of these the late flower showed higher RPKM value in the correspondence of CO transcript level. Down regulation of *COL9* (*MK098126, GO:0005634*) was observed in all wild-growing samples. Since overexpression of *COL9* also

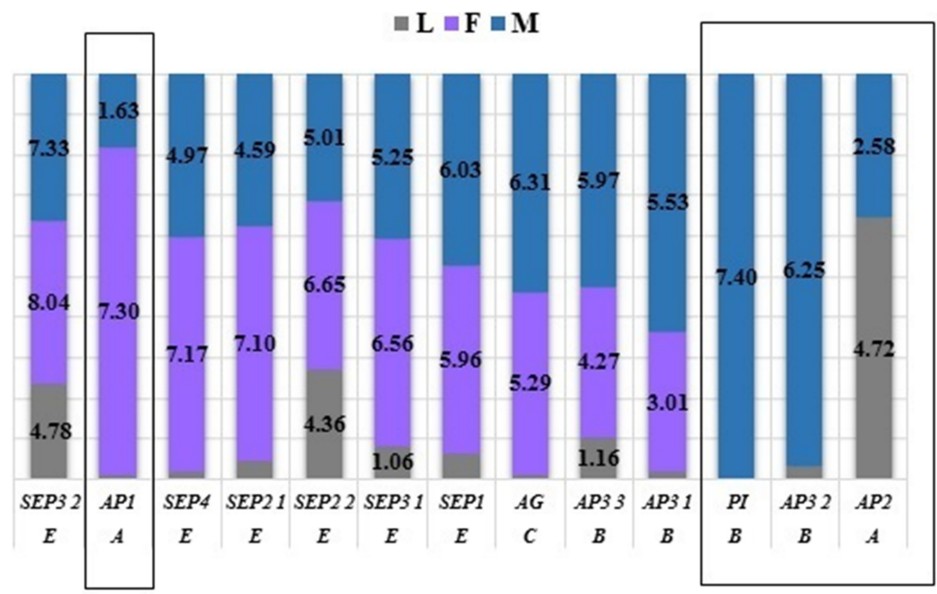

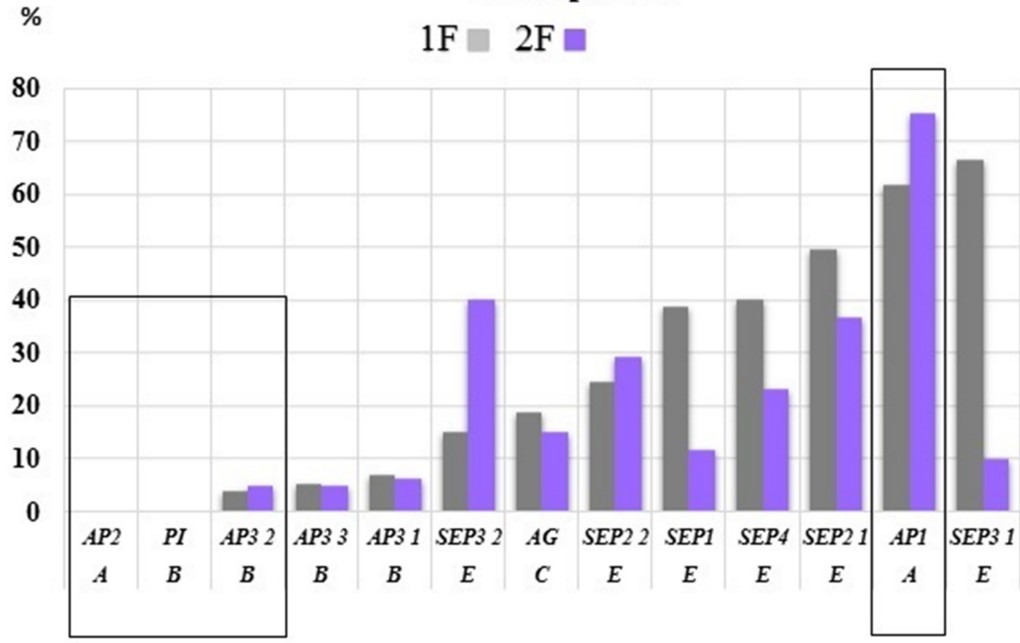

**Figure 8 Expression differences between ABCE genes.** Normalized expression rate were determined by using Trinotate (A) and RPKM (B). (A) The highest expression of *AP1* were confirmed in female flowers (F) and exclusive expression of *PI* and *AP3* in male flowers (M). (B) During female flower development the same trend might be observed. The male characteristic *AP2* and *PI* were not expressed, hovever *AP3 2* showed minimal expression. The highest expression of *AP1* were detected in both 1F and 2F phenophases.                                 

delayed flowering by reducing expression of CO and FT (*Cheng & Wang, 2005*), our results confirm the natural circadian rhythm of wild growing samples. Over-expression of *COL5* can induce flowering in short-day grown *Arabidopsis* (*Hassidim et al., 2009*). We observed the higher *COL5* expression in female flowers, mainly in the case of in vitro plants where pistillate flowering showed intensive growth under short light conditions.

The next most characteristic transcript was an *ILR3* (*MK098117, GO:0046983*) homolog of *IAA-LEUCINE RESISTANT 3* protein representing the helix-loop-helix protein family. *ILR3* is required to maintain Fe homeostasis in correlation with a large amount of $Fe^{3+}$ observed in the sepals in *A. thaliana* (*Zhang et al., 2015*). Since this transcript was found to be expressed roughly equally in all investigated samples (1F, 2F, 1M, 2M, M, and L), approximately, we assumed that to keep at appropriate level the iron homeostasis is essential to all flower types in fact to maintain hormonal and redox balance for fertility (*Sudre et al., 2013*).

Among the hormone related factors the auxin responsive AUX/IAA protein family genes such as *IAA27* (*MK098121, GO:0005634*) and *IAA9* (*MK098120, GO:0005634*) were found to be characteristic in both reproductive tissues. These transcriptional factors are repressors of early auxin response genes at low auxin concentrations (*Liscum & Reed, 2002*). RT-qPCR and in situ hybridization have shown that tissue-specific gradient of *IAA9* expression which was established during flower development, and the release of which, upon pollination and fertilization, triggers the initiation of fruit development (*Wang et al., 2009*). These results explain the higher expression rate in female tissues in both early and late phenophases in *Ambrosia*. The auxin efflux carrier homolog, *PIN-FORMED 1* (*PIN1, MK098122, GO:0009734*) was also found to be upregulated in early female flowers (1F) indicating the female tissue determination might be under strong control of auxin. These results correspond to the study of *Ceccato et al. (2013)* where *PIN1* expression and cellular localization in the ovule were essential for correct auxin efflux in early stages of female gametophyte development. In the ethylene response pathway the protein *ETHYLENE INSENSITIVE3* (*EIN3, MK098123, GO:0003677*) regulates ethylene responsive genes as a positive regulator and results in flowering delay via repression of the floral meristem-identity genes *LFY* and *SOC1* (*Achard et al., 2007*). In our flowering system it was upregulated in M, 1M, 2M libraries resulting no expression of *SOC1* and *LFY* in male tissues (see Figs. 3 and 4). No *LFY* homolog was identified in any of the floral libraries in corellation of higher expression of this TF. Wild growing *Ambrosia* individuals grow male flowers intensively in late summer, which is a delayed state compared to the female blossom. However, *SOC1* transcript was found only in the in vitro samples referring to the altered (earlier) floral regulation we assumed that this fenomena is based on changes in hormonal pathways.

NAC-domain containing TFs are essential for normal plant morphogenesis which representatives are the *CUP-SHAPED COTYLEDON* proteins encoded by *CUC1* (*MK098118, GO:0003677*) and *CUC2* (*MK098119, GO:0003677*) genes less expressed in our flower samples. Mutations in *Arabidopsis CUCl* and *CUC2* caused fusion of cotyledons, sepals, and stamens suggesting that these factors are responsible for mechanisms that separates organ developing at adjacent positions (*Aida et al., 1997*). No expression of *CUC1*

was detected in wild growing pistillate tissues and its expression was lowest among the investigated TFs in staminate samples. In in vitro samples *CUC2* was expressed only in early phenotypes (1F) in female flowers with a minimal expression rate. Since these factors are involved mainly in early development of cotyledonary primordia (*Aida et al., 1997*), the phenomena of downregulation of these genes may be explained by the flowering separation that has overtaken this critical state in our samples.

The homeodomain like MYB transcription factors are involved in the control of the cell cycle of plants. The majority of them are miRNA regulated and are involved in numerous cell biochemical pathways like hormonal regulation. The GA pathway related *MYB33* (*MK098116, GO:0003677*) homolog was expressed only in male flower in wild growing *Ambrosia* samples in a correspondence with *Rocheta et al. (2014)* and *Millar & Gubler (2005)* in which studies the *GAMYB-Like MYB33* was proved to facilitate anther development redundantly. In in vitro originated 1F and 2F libraries homolog transcripts of *MYB33* were also found indicating repeatedly altered hormonal regulation of this system.

## Expression of female flower specific genes

As expected, all of these investigated genes are not expressed in M, L, 1M, and 2M libraries (RPKM = 0) except of *BEE1* in leaf samples with a very low value. In early female flower development the serine/threonine protein kinase *PBL9* (*MK098095, GO:0004674*) showed an exceptionally high value. *SUP, TCP12*, and *MYB5* were expressed two times greater than in matured flowers before the ovule formation. Two catalytic protein genes *LAC2* and *ACA7* were also found as unique genes in late 2F samples with low expression suggesting a not characteristic role, but taking part in necessary metabolic processes during floral maturation

*PBL9 (alternatively APK1)* is very weakly expressed in flower tissues. It was described as *Arabidopsis* thaliana serine/threonine protein kinase that phosphorilates tyrosine, serine and threonine *(APK1)* during signal transduction (*Hirayama & Oka, 1992*). The *APK* family is negatively regulated by the floral homeotic protein AG *(AGAMOUS)* (*Ito et al., 1997*) involved in the control of organ identity, which expression was equal in early and late female libraries, respectively. Unique expression of *PBL9* in female flowers indicated that the development of staminate and pistillate tissues are regulated by different kinases at protein level during signal transduction mechanisms. Similarly, a receptor-like protein kinase was also reported as female specific gene in the monoecious *Q. suber* (*Rocheta et al., 2014*). Additionally, different expression of these gene showed altered protein phosphorilation during female flower formation, accurately it is more significant at the beginning of the pistillate flower development.

Despite of SUP is not a sex determination gene (not located on the sex—Y—chromosome) it was considered as female specific gene. Their function in female flower differentiation pathway was proved in dioceious plant *Silene latifolia*. During the development of the female flower in *Silene latifolia*, the expression of SUP is firstly detectable in whorls 2 and 3 when the normal expression pattern of the B-class flowering genes was already established and persisted in the stamen primordia until the ovule had matured (*Bowman et al.,*

*1992*; *Kazama et al., 2009*). It is probably controlled by sex determination genes on the Y chromosome in dioecious plants.

The role of MYB transcription factor superfamily in plant reproductive development is unequivocal. *MYB* proteins include a conserved domain, the *MYB* DNA-binding domain regulating a variety of plant-specific processes and they may play roles in different plant species (*Ambawat et al., 2013*). For example *R2R3 MYB*, and *MYB88* regulates female flower reproduction in *A. thaliana* (*Makkena et al., 2012*), *MYB35 (MSE)* regulates male specific expression in *Asparagus officinalis* (*Murase et al., 2017*). *MYB1* and *MYB16* regulates petal development in *Petunia hybrida* and *A. thaliana* (*Baumann et al., 2007*; *Noda et al., 1994*). Overexpression of *MYB61* was demonstrated in several aspects of plant growth and development, such as xylem formation and xylem cell differentiation, and lateral root formation (*Arsovski et al., 2009*; *Liang et al., 2005*; *Matías-Hernández et al., 2017*; *Romano et al., 2012*). Importance in seed coat formation, trichome morphogenesis and mucilage secretion of *MYB5* and *MYB61* was also described by several studies (*Arsovski et al., 2009*; *Gonzalez et al., 2009*; *Li et al., 2009*; *Liu, Jun & Dixon, 2014*) indicating overexpression in seed coat and siliques. In our investigations *MYB5* and *MYB61* showed unique expression in female flowers, however, *MYB5* (*MK098099*) *MYB61* (*MK098101*) was found to be more characteristic in early and late phenophases, respectively.

*TCP12* belongs to the plant-specific TF family TCPs. The founding members of this family have been characterized to be involved in growth, cell proliferation, and organ identity in plants (*Yang et al., 2012*). TCP1-like genes are involved in floral development with different regulatory and evolutionary mechanisms underlying the diverse forms of floral symmetry, such as dorsal/ventral dosage effect (*Busch et al., 2011*). We found that the female specific *TCP12* was stronger expressed in earlier phenophases suggesting a *TCP12*-related zygomorphic regulation at the beginning of floral morphogenesis in this species.

The transcription factor *BEE1* is a positive regulator of brassinosteroid signaling (BR) regulating plant development and physiology. The BR signal is transduced by a receptor kinase-mediated signal transduction pathway which ultimately results in altered expression of numerous genes. During flowering brassinosteroids stimulate floral transition through positively regulating the circadian clock pathway genes and inhibiting *FLC* repressor (*Domagalska et al., 2007*; *Zhu, Sae-Seaw & Wang, 2013*). Deficiency of this phytohormones lead to the delay of flowering and because of the disruption of the pollen tube, BR deficient plants are practically male-sterile (*Clouse, 1996*). Characteristic overexpression of the BR enchancer *BEE1* uniquely in late-female flowers suggest on one hand an altered regulation of BR signal transduction than in male flower development, on the other hand it confirms an enhanced BR mediated floral transition related to circadian clock in *Ambrosia artemisiifolia*.

Transcription factor *WIP2* is a zink finger protein and was found to be responsible for transmitting tract formation in the gynoecium, which is important for pollen tube growth (*Crawford & Yanofsky, 2008*). *WIP2* plays role in replum development and regulation of *AGL8/FUL* (*MK098076, GO:0000957*), which is required for normal pattern of cell

division, expansion and differentiation during morphogenesis of the silique (*Marsch-Martínez et al., 2014*). Expression of this gene was equal in early and late developmental flowers indicating an intensive cell division in the gynoecial tissues during fruit development.

*LAC2* is a multicopper-containing glycoprotein belonging to the laccase family. The exact function of *LAC* genes is not yet cleared. Laccase genes were highly expressed in lignifying tissues and are known to be associated with lignin synthesis (*Gavnholt & Larsen, 2002*; *Sato et al., 2001*). In *Arabidopsis* array experiments, the majority of investigated laccase members were found primarily in root tissues and in flower and stem samples secondly. In a mutant screening experiment, *LAC8* and *LAC2* deficient mutants lead to early flowering and reduced root elongation phenotypes. *LAC2* expression was also detected in flower tissues in the *Arabidopsis* array (TAIR database) (*Gavnholt & Larsen, 2002*). In our *Ambrosia* flowering system, LAC2 was found to have been expressed uniquely in late 2F flowers; however, it was present with a reduced value of order compared to the other female-specific genes in this library.

The role of *ACA7* in flowering pathways is not yet reported. It belongs to the Zn metalloenzymes carbonic anhydrases (CAs) that catalyze the reaction of reversible hydration of $CO_2$ with exceptionally high efficiency. CAs have been involved in a broad range of biochemical processes that involve carboxylation or decarboxylation reactions including photosynthesis and respiration. CAs also participate in pH regulation, inorganic carbon transport, ion transport, water, and electrolyte balance (*Henry, 1996*; *Smith & Ferry, 2000*; *Tashian, 1989*). However, an interesting correlation revealed, that *ACA7* location was mapped on the same chromosome region as *EARLY FLOWERING 3 (ELF3)* including the earliness per se locus Eps-Am1 in diploid wheat (*Triticum monococcum*). This locus affects the duration of early developmental phases and it is responsible for the optimal photoperiod sensing and vernalization during flowering time (*Gavnholt & Larsen, 2002*). Due to *ACA7* location it may be considered that carbon-based catalytic processes are strongly related to *ACA7* in female flower development in *Ambrosia artemisiifolia*. These results suggest that female flower differentiation may differ mainly in regulatory elements of the flowering pathway genes such as altered signal transduction and TF activity.

## Expression of male flower specific genes

Of the annotable genes, the mostly expressed sequences showed homology with the followings:

The homolog of desiccation-related protein, *PCC13-62* (*MK098104, GO:no terms*) showed the highest RPKM value, 1,126.37, which is activated during the drought stress in the pollen, seeds, and vegetative organs in succulent plants (*Giarola et al., 2018*). It was expressed exclusively in the stylopodium of the hermaphrodite flower in *Mucona sempervivens*, where the nectary is located (*Zha et al., 2013*). Since the male flower pistillodium is a modified pistil with a function to help dispensing the pollen from staminate flowers in *Ambrosia artemisiifolia* (*Payne, 1963*), the anatomical presence of this transcript may be explained. Functionally, however, this protein may serve to protect the

cytoplasmic contents in the pollen, since the pollen grain is an extremely desiccated structure at maturity. Desiccation-related proteins were also reported in the *A. thaliana* columbia ecotype pollen (*Sheoran et al., 2006*).

The second most significantly expressed sequence was the *CYP450 86B1* (*MK098105, GO:0004497*) homolog. This gene was found to be expressed in many parts of the flower, including epidermis, anther, and pistil. In young inflorescences, this gene effects the suberin and cutin biosynthesis playing an important role in defensive mechanisms during biotic and abiotic stresses and supports the maintenance of reproduction (*Compagnon et al., 2009*; *Duan & Schuler, 2005*).

The third most significantly expressed gene homolog was found from the lipolytic enzyme family, *GDSL2* (*MK098106, GO:0016021*), which was expressed during petal differentiation and is responsible for lipid catabolic processes (*Ji et al., 2017*). Ji et al. identified a *GDSL* lipase gene in *Brassica rapa* playing an important role in the anther and pollen development. *Arabidopsis* GDSL lipase 2 plays role in pathogen defense via negative regulation of auxin signaling and a positive correlation between late flowering and resistance to *Fusarium oxysporum* in *Arabidopsis thaliana* natural ecotypes was observed related to GDSL lipases (*Lyons et al., 2015*).

The above described male characteristic genes with outstanding high RPKM values refer to the most intensive physiological processes that are related to pollen exposure and stability and biotic and abiotic defense during anther formation.

Because all of the most male characteristic genes are related to flowering, secondary metabolism, plant cell structure, biotic and abiotic stresses, these may be considered to be essential to male specificity in the common ragweed. Gene homologs discussed below were found to be important for male flower physiology based on RPKM values. *TET8* (*MK098111, GO:0016021*), which encodes a transmembrane protein tetraspanin 8, member of tetraspanin family. TET8 play important role in plant development, reproduction and stress responses (*Reimann, Kost & Dettmer, 2017*). Expression of tetraspanins including tetraspanin 8 were described during the pollen development and in mature pollen in *Arabidopsis* thaliana (*Boavida et al., 2013*; *Honys & Twell, 2004*; *Pina et al., 2005*). *pMADS2* was found expressed in flowers in the second whorl, however in vegetative organs were not found in *Petunia* (*Van Der Krol et al., 1993*). In *Ambrosia*, this gene was found to be homologous to PI that is predominantly expressed in petals and stamens, and less in carpels and sepals. The 3-oxoacyl-(acyl-carrier-protein) synthase coding *OAS* (*MK098107, GO:0006633*) is responsible for the condensation reaction of fatty acid synthesis and confers resistance to low temperatures by maintaining chloroplast membrane integrity (*Dayan & Duke, 2014*). This protein is also involved in the regulation of fatty acid ratio during seed metabolism. It is required for embryo development, especially at the transition from globular to heart stage (*Hakozaki et al., 2008*). Another gene family responsible for membrane structure, the syntaxins (SYN) showed also unique expression in male tissues. These proteins play role in fusion of transport vesicles to target membranes. Inactivation of genes from two syntaxin families has led to lethality of the male gametophyte (*Sanderfoot et al., 2001*). *NIP 4 1* and *NIP 4 2* are pollen specific aquaporins playing role in pollen germination and pollination in *A. thaliana*. These genes

are essential to the growth of pollen tubes and were considered to be exclusive components of the reproductive apparatus of angiosperms with partially redundant roles in pollen development and pollination (*Di Giorgio et al., 2016*).

MYB proteins are a superfamily of TFs that play regulatory roles of gene expression controlling direct organ development and defense responses in plants. We found uniquely expressed MYB transcription factors related to flower development in male tissues. *MYB80* (*MK098109, GO:0003677*) showed the highest expression in male tissues playing an essential role during anther development and production of tapetum and microspores (*Higginson, Li & Parish, 2003*) (*Phan et al., 2011*). *MYB26* (*MK098115, GO:0003677*) regulates lignified secondary cell wall thickening of the anther endotechium, which is necessary for anther dehiscence. It may play role in specifying early endothecial cell development by regulating a number of genes linked to secondary thickening (*Mitsuda et al., 2006*; *Yang et al., 2007*). *MYB35* (*MK098110, GO:0003677*) is required for anther development and early tapetal function during microspore maturation. It regulates callose dissolution and is required for microspores release from the tetrads (*Zhu et al., 2011*). The *MYB44* (*MK098112, GO:0003677*) represses the expression of protein phosphatases 2C in response to abscisic acid (ABA). It confers resistance to abiotic stresses dependent of ABA. The overexpression of *AtMYB44* enhanced stomatal closure to confer abiotic stress tolerance in transgenic *Arabidopsis* (*Jung et al., 2008*).

## CONCLUSIONS

Genetic program of formation of separated male and female flower development in monoecious plants is a less understood mechanism in flowering biology. Available genomic data of the majority of these species are insufficient or inexistent to date. Another difficulty is that the reproduction biology of *Ambrosia artemisiifolia* is less studied and the majority of information has been revealed exclusively in the pollen formation as part of male flowering biology. Today, large DNA or RNA data sets of non-model organism may be attainable using NGS technology; therefore, we sequenced and compared transcriptome libraries of wild growing male, female, leaf and in vitro female flowers representing early and late developmental phases. Comparative studies revealed a subset of transcripts that were DE in the different libraries known in flower or plant development in *Arabidopsis* thaliana. However, genes showing differential expression previously were not characterized in *Ambrosia artemisiifolia* during flowering.

Genome-wide transcriptional profiling in five libraries revealed a surprising number of transcripts that were DE playing role—non exclusively—in plant or flower development, but also in signal transduction, redox and abiotic stress mechanisms. Induction of floral meristem initiation is proceeded by a complex regulation network just in the vegetative phase which regulatory elements are the so called flowering pathway genes. Based on expression pattern of these pathways we demonstrated that the common ragweed belongs to the short-day plants and flowering induction depends on *PHYB*-related light signals. Because of the small amount of collectable pistillate flowers, plants were also in vitro cultivated on meta-topolin supplemented media. In this way, influence of

hormonal and photoperiodic pathways were modeled on the flowering pathway genes. Based on gene expression differences of flowering pathways in male and female samples and the shifts to intensive female morphogenesis under in vitro cultivation conditions, we concluded that before the generative transition, the determination of floral gender takes place just in vegetative phase during the vegetative pathway dominancy that defines the subsequent floral organ morphogenesis. Transition to *AP1* and FOIGs induction is leaded by hormonal and photoperiodic pathways depending on which dominancy the female or male flower initiation come into being. Thus, flower identity is decided just in vegetative growth. Investigation of male flower showed that photoperiodic pathway induce the generative transition through FT/FD complex; however, GA-related *SOC1* expression was not observed indicating it has no role in male gender formation. On the other hand, expression of FT/FD was lacking in in vitro female samples that was favorable to *SVP* (*MK098077*, *GO:0000977*) or *AGL24*. Transcripts of *LFY* homolog were not found in any of the libraries, therefore it appears likely that floral organ development is induced through *LMI–CAL* regulation in female. However, *FUL* and directly *(AP1)—AP2* regulation in male flowers in only induced when the *AP1* repression by *TFL1* was also experienced. Expression analysis of TFs confirmed the greatest importance in regulation of the circadian clock taking part in progress or delay of flowering. The followings are in order the hormonal-redox balance to maintain fertility and the structural and morphological regulations during *Ambrosia artemisiifolia* reproduction.

## ACKNOWLEDGEMENTS

We express our thanks to our retired colleague Dr. Csaba Pintér for the pictures in the Fig. 1.

### Funding

The research leading to these results has received funding from the European Union under Horizon-2020-SFS-2015-2 Programme grant agreement No. 677407 (SOILCARE project, https://www.soilcare-project.eu/). This work was supported by the GINOP-2.3.2-15-2016-00054 project. Additional support was received from: the New National Excellence Program of the Ministry of Human Capacities (ÚNKP-2016-4-IV, ÚNKP2017-4-IV-PE-1). The funders had no role in study design, data collection and analysis, decision to publish, or preparation of the manuscript.

### Grant Disclosures

The following grant information was disclosed by the authors:
European Union under Horizon-2020-SFS-2015-2 Programme: 677407 (SOILCARE project).
GINOP-2.3.2- 15-2016-00054.
New National Excellence Program of the Ministry of Human Capacities:
ÚNKP-2016-4-IV, ÚNKP-2017-4-IV-PE-1.

## Competing Interests

The authors declare that they have no competing interests.

## Author Contributions

- Kinga Klára Mátyás conceived and designed the experiments, performed the experiments, analyzed the data, prepared figures and/or tables, authored or reviewed drafts of the paper, performed experiments and data analysis.
- Géza Hegedűs analyzed the data, contributed reagents/materials/analysis tools, prepared figures and/or tables, statistical analysis,bioinformatics, software development and database managment.
- János Taller conceived and designed the experiments, contributed reagents/materials/analysis tools, consevied and designed the experiments and acquired funding.
- Eszter Farkas preparation of plant material for experiments.
- Kincső Decsi preparation of plant material for experiments.
- Barbara Kutasy preparation of plant material for experiments.
- Nikoletta Kálmán contributed reagents/materials/analysis tools, performed RT-qPCR analysis.
- Erzsébet Nagy preparation of plant material for experiments.
- Balázs Kolics prepared figures and/or tables, performed and designed PCR experiments.
- Eszter Virág conceived and designed the experiments, performed the experiments, analyzed the data, contributed reagents/materials/analysis tools, prepared figures and/or tables, authored or reviewed drafts of the paper, approved the final draft, supervised the study, performed bioinformatics, data analysis and wrote the manuscript.

## DNA Deposition

The following information was supplied regarding the deposition of DNA sequences:

The 80 flowering responsible genes (mRNA sequences) are described here are accessible via GenBank accession numbers MK098047 to MK098126.

## Data Availability

The raw data is available in the Supplemental Files: the raw data file of SQL database (Dataset S1); Dataset S2 includes CDS sequences of 80 genes; and the third raw data files of three male and three female local blast libraries of the investigated genes (Dataset S3).

## Supplemental Information

Supplemental information for this article can be found online at http://dx.doi.org/10.7717/peerj.7421#supplemental-information.

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
