# Peer review of "Different expression pattern of flowering pathway genes contribute to male or female organ development during floral transition in the monoecious weed Ambrosia artemisiifolia L. (Asteraceae)"

_PeerJ, doi:10.7717/peerj.7421_

## Round 0.1 · original submission · Major Revisions

Dear author

Your paper has been assessed by two reviewers and myself as academic Editor.

As you could see below, the manuscript needs a major revision. Most importantly: Please clarify the statistical significance of the male library.

Please address most concerns of the reviewers and submit a revised version of the manuscript. Please include a detailed response to each reviewer.

·

Basic reporting

The English is clear.
The references need a minor changes.
The structure of the article is appropriate but figures needs a revision

Experimental design

The research question is well defined and is relevant and the investigation was rigorously conducted but needs to be more detailed in some aspects referred in the general comments

Validity of the findings

no comment

Additional comments

The paper proposed by Mátyás et al, provides an overview of the genes expression patters along flower development of Ambrosia artemisiifolia. To accomplish this, authors refer to libraries prepared and published on a previous data report (doi: 10.3389/fpls.2016.01506), where RNA-Seq was performed to Male and Female flowers as well as leaves. The work proposed provides the follow up to the data report. Here, two more female samples were obtained and sequenced. The five data sets were analyzed together, providing an extensive, detailed, focused and clear overview of patter expression. Also, authors have carefully detailed the introduction, which represents one of the most interesting, complete and easy to read state-of-the-art, in flower development field. It should also be noted that the detailed Results & Discussion section is very clear and well structured, providing a very easy access to the information stored under this work.

Along this manuscript, major and minor questions have come to surface, that should be clarified or corrected by the authors, before final publication.

Major questions
1. Under this study, five libraries are compared and, under Results and Discussion, there are segments where Female (and Male) exclusive transcripts are accessed, although, there are 3 Female libraries and only one Male library. I am concerned relating to the statistical significance of the Male library under this context. Is one single library relevant to provide a complete Male Knowledge, whereas it was need 3 females to provide that? I wonder if the addiction of more male libraries does not change the results.
2. The number of individuals used along sampling is not detailed. How many individuals were used along sampling? Are 1F and 2F from the same individual(s)? How many buds were collected from each one of the nine developmental stages? If multiple buds were used, was it equally distributed? Also, according to repositories, there are 2 runs for 1F and 2 for 2F. How each one of these runs differ from each other? Are they biological or technical replicates?
3. Authors used both Bowtie and Bowtie2 on this analysis, each one, were used with different instruments. These programs have different standards, scopes, defaults, etc. I wonder why did authors decided to use both that increment the entropy of the analysis. Also, as standard and defaults are different, authors need to clearly detail which parameters were used and the respective values. Considering that all samples have more than 50bp reads, bowtie2 should had been used on all samples. Authors should justify the option made.
4. Along Methods, Line 258, under «Construction of A. artemisiifolia transcript datasets (...)» authors state that «The resulting differentially expressed transcripts were further clustered according to their expression patterns by applying SQL Database Manager». This statement should be more detailed: (a) latter, on results and discussion, authors clarify that Microsoft SQL Server was used as Database Management System. This information should be provided early in Methods. (b) There is no information of which kind of data was stored under this database. As far as I can understand, this database only contained the expression for each gene, but that is not stated. (c) The database construction should be detailed as supplement material. A different choose of fields along table creation may lead to different results, as computers will interpret the data diferentially. (d) The interaction mechanism with this database needs to be clarified. Did authors executed SQL queries directly on the database using a SQL client (in that case, which one?) or it was used any programming language interface, as R, Python? (e) Is this database publicly available? If so, provide instructions to access.
5. The usage of RPKM/FPKM is broadly spread along RNA-Seq studies and the usage of this method to access gene abundance is well established. Although, bioinformatics’ communities are discussing the application of this calculation methodologies when comparing different data sets and studies. In order to better compare the expression along different studies, it is more acceptable the usage of TPM (Transcripts Per Kilobase Million). As the authors are using reads obtained from different instruments, I wonder if the calculation method is the best one.
6. The proposed article is very extensive and detailed as already stated on this review introduction. Nevertheless, as authors have five libraries to work on, I wonder, if a cis/trans or co-expression analysis could be performed. Authors may compare the expression patterns of all the identified genes and check which groups are co-expressed/co-repressed, which may indicate co-regulation of those genes. I do not think that this analysis is fundamental to this paper, but it could contribute to the overall knowledge of A. artemisiifolia regulatory networks.
7. Pp 304. Authors state that for RT-qPCR analysis only one reference gene was used. Usually for more robust results two or three reference genes are used when analyzing relative expression.

Minor questions
1. Along Materials and Methods, on the «Plant materials and cultivation conditions», authors present the sampling first and the cultivation on the end. As this steps were taken in the opposite order, authors may want to adapt the text of this segment and start by describing the cultivation process and then the sampling methodology.
2. The standard protocol of Illumina RNA-Sequencing states that cDNA is constructed using olido(dt) primers. Authors may include information on «RNA-seq library construction and sequencing» segment, to inform if cDNA for RNA-Sequencing was produced using oligo(dt)s or random hexamers.
3. On titles, «A. artemisiifolia» is not in italics.
4. RSEM package should be better cited (please, refer to https://doi.org/10.1186/1471-2105-12-323).
5. Under «Digital gene expression analysis» it is refereed that RPKM was calculated. Although the information provided on the manuscript and on the repositories is that the authors performed paired-end sequencing. If both forward and reverse reads data sets were added to the programs, the authors did not calculated the RPKM but the FPKM instead (Fragments per kilobase per million mapped read). As the formula used is the same, this difference is only semantics but should be corrected in order to avoid confusing the reader. If the programs were not used with both Forward and Reverse reads at the same time, that should be clarified and justified as well.
6. Primer3 should be better cited. This software is released with version control and so, the version used should be cited (check the top of the webpage). Also, articles should be cited instead of the webadress (please refer to 10.1093/nar/gks596 and 10.1093/bioinformatics/btm091).
7. Authors used non-redundant protein database from NCBI and correctly provide the respective link, although, as this database is not curated and is not released with version control, I would recommend authors to provide the download date.
8. Pp 344, 632 – Reference format. Should be Rocheta M et al., 2014b
9. BLASTx is part of a package of BLAST programs and should be cited properly. Please, refer to blastx documentation to identify the correct paper to cite. Is none is specifies, authors should cite BLAST+ package, by referring to its proper article(s).
10. On line 362: «(2*100 bp paired-end)». The information is redundant. Authors should state «(2*100 bp)» or «(100 bp on paired-end sequencing)». The first one is most commonly found.
11. On «RNA sequencing and assembly of A. artemisiifolia floral libraries» segment, authors should include the reads length for the new libraries (1F and 2F). I found on supplementary material that those reads have 80 bp, but that information should be stated on the main manuscript as it the base of all the provided results.
12. On lines 370-371: «Their de novo assembly resulted in 109,452 (1F) and 97,239 (2F)». Please add «contigs» to the end of this sentence as «Their de novo assembly resulted in 109,452 (1F) and 97,239 (2F) contigs». It is not clear if you are talking about contigs, scaffold, etc if we do not cross with supplementary material.
13. I would like to propose to authors the utilization of «de novo» in italics, as that is a Latin expression and that is generally the standard. Nevertheless, I noted that authors were consistent by do not use italic on this expression.
14. Figures numbering needs to be completely revised. The order of calling figures is: 1, 2, 7, … Also, figure 5 is never called on text.
15. On line 400-401: «Representation of the significant sub-ontologies of cellular component, molecular function, and biological process are summarized in the Supplementary Figure 2». Do authors want to call Supplementary Figure 1 instead? I have only found one Supplementary Figure.
16. Figure 4: The «A» letter used to identify panel A is on top of the plot yy scales.
17. Pp 429-430. Authors use Q-RT-PCR in the subtitle and in text RT-qPCR. Please make it uniform.
18. Pp 822 – garfical or graphical?
19. Table 1. There are two Rocheta M references used in this manuscript, however, in table 1 it only shows Rocheta M et al., 2014. It is a or b?

·

Basic reporting

The basic reporting style is adequate.

Experimental design

Based on the description of the experimental design, there is no biological replication. Certainly there was none for the RNAseq portion of the study, and it is unclear if there was biological replication for the qRT-PCR protion of the study since there was no mention of the number of biological replicates collected at each floral developmental stage in the first paragraph of the methods. As such, no differential expression of genes should be reported.

Additionally, although there is a reference noting the use of the internal reference gene used for qRT-PCR, there is no solid evidence to suggest that that gene is not itself differentially-expressed through floral development. This could be easily remedied (assuming replication of the RNAseq libraries is done) by specifically showing the expression pattern of the GAPDH gene in the RNAseq studies (which are generally more accurate than qRT-PCR in any case).

However, this data does allow the presentation of the sequences of key flowering genes that were assembled from the sequenced libraries.

Validity of the findings

Sequences of floral genes are likely accurate. None of the differential expression is valid.

Additional comments

This reviewer understands the difficulty and cost of running RNAseq studies on multiple samples with adequate replication. However, the cost and time is not justification for failing to perform adequate replication.

If this paper was rewritten to just examine the sequences of floral genes (or indeed just the sequences of all genes expressed in the floral organs- GO ontology, any qualitative differences in sequences between floral genes in male and female organs, etc. then that would be acceptable for publication. Likewise, if there was true biological replication and subsequent qRT-PCR analysis of specific floral genes, then that too would be publishable- assuming there was proof provided that the internal control gene was stably expressed in the floral organs through development.

---

## Round 0.2 · Minor Revisions

Dear author,

Your paper has been assessed by one reviewer and myself as academic Editor.

As you could see below, the manuscript has improved considerably, but the reviewer have not yet given green light of full acceptance, but still have some minor comments. I think you can easily and rapidly incorporate them.

Please address the minor issues and submit a revised version of the manuscript.

·

Basic reporting

Minor questions
1. On titles, «A. artemisiifolia» is not in italics. Still missing in line 244
2. BLASTx reference is still missing in line 334.
3. «Pp 429-430. Authors use Q-RT-PCR in the subtitle and in text RT-qPCR. Please make it uniform.» Now, I find 4 forms:
1. qRT-PCR: Lines 415 and 507
2. RT-QPCR: Lines 34, 414 and 496
3. RT-qPCR: Lines 222, 298 and 615; Figure 6

Experimental design

No comments. It is ok.

Validity of the findings

No comments. It is ok.

Additional comments

Just revise the manuscript for minor corrections.

---

## Round 0.3 · Minor Revisions

Dear author

I can read that you have addressed all the reviewers concerns. The reviewers comments have been responded adequately and your manuscript has improved considerably. Nevertheless, the section editor Gerard Lazo has suggested some further editions. His comments are as follows and I think you can incorporate them in order to improve the quality of your work. Please address these issues and submit a revised version of the manuscript.

Comments from Gerard Lazo (Section Editor): "As I reviewed the manuscript I was disappointed to note the lack of annotation development with the key genes involved in the flowering process. It would go a long way if the authors were to add this element to the manuscript. The GO annotations listed in Supplemental Figure S1 only have the generic description, and there is no connection to any of the tangible sequence data mentioned. There are sequences described in the declarations as being available from Genbank as MK098047 to MK098126, but there is no mention of them in the manuscript. In addition, the sequences within supplement S2 file are poorly represented; being in non-standard FASTA-like format and having essentially no annotation. A question would be how these sequences relate to those mentioned at the NCBI Genbank site.

The manner in which the SQL data was deposited left no instruction for uploading into the Microsoft product, and each of the files were in binary form. To make the data more broadly available, a simple sequence dump into a textual *.txt data files and *.sql schema files would be more appropriate for the dataset S1 included data file; some sort of README file should accompany it.

Journal manuscripts are often scanned by text-mining software that locates and extracts core data elements, like gene function. Adding standard ontology terms, such as the Gene Ontology (GO, geneontology.org) or others from the OBO foundry (obofoundry.org) can enhance the recognition of your contribution and description. This will also make human curation of literature easier and more accurate. Though, as mentioned, there were no connections between available terms and the sequence data.

In general, this was a data-rich manuscript and so if these issues can be addressed, the manuscript is well on its way to being ACCEPTED

Additional suggestions:
EDITS LINE NO: / BEFORE / AFTER / [COMMENTS]
LINE 31: / on / on the / [.]
LINE 32: / expressions / expression profiles / [.]
LINE 33: / genes was / select genes were / [.]
LINE 35: / against / respond to / [.]
LINE 36: / High gene expression of homologous such as / Increased levels of gene expression for homologues in each tissue type of / [.]
LINE 37: / Homologues / Homologue / [.] LINE 39: / by TFL1 homolog / by the TFL1 homolog / [.]
LINE 63: / invasion pathways, / invasion migratory pathways, / [.]
LINE 63: / there are at high risk / there is high risk / [.]
LINE 79: / Functional / Functionally / [.]
LINE 110: / Such way may / Such a way may / [.] LINE 111: / . / . / [Stopped editing.]"

---

## Round 0.4 · accepted · Accept

Dear author

I can read that you have addressed the editorial concerns. The comments have been responded adequately.

I congratulate you for the nice piece of work, which will add value to PeerJ.